# Drought effects on leaf fall, leaf flushing and stem growth in the Amazon forest; reconciling remote sensing data and field observations

Thomas Janssen[1], Ype van der Velde[1], Florian Hofhansl[2], Sebastiaan Luyssaert[3], Kim Naudts[1], Bart Driessen[4], Katrin Fleischer[5] and Han Dolman[1]

[1] Department of Earth Sciences, Vrije Universiteit Amsterdam, Amsterdam, the Netherlands
[2] International Institute for Applied Systems Analysis (IIASA), Laxenburg, Austria
[3] Department of Ecological Sciences, Vrije Universiteit Amsterdam, Amsterdam, the Netherlands
[4] Department of Computer Science, Universidad de Alcala de Henares, Madrid, Spain
[5] Department of Biogeochemical Signals, Max Planck Institute for Biogeochemistry, Jena, Germany

*Correspondence to*: Thomas Janssen (t.a.j.janssen@vu.nl)

**Abstract.**

Large amounts of carbon flow through tropical ecosystems every year, from which a part is sequestered in biomass through tree growth. However, the effects of ongoing warming and drying on tree growth and carbon sequestration in tropical forest is still highly uncertain. Field observations are sparse and limited to a few sites while remote sensing analysis shows diverging growth responses to past droughts that cannot be interpreted with confidence. To reconcile data from field observations and remote sensing, we collated *in situ* measurements of stem growth and leaf litterfall from inventory plots across the Amazon region and other Neotropical ecosystems. This data was used to train two machine learning models and to evaluate model performance on reproducing stem growth and litterfall rates. The models utilized multiple climatological variables and other geospatial datasets (terrain, soil and vegetation properties) as explanatory variables. The output consisted of monthly estimates of leaf litterfall ($R^2 = 0.71$, NRMSE = 9.4%) and stem growth ($R^2 = 0.54$, NRMSE = 10.6%) across the neotropics from 1982 to 2019 at a high spatial resolution (0.1°). Modelled time series allow to assess the impacts of the 2005 and 2015 droughts in the Amazon basin on regional scales. The more severe 2015 drought was estimated to have caused widespread declines in stem growth (-1.8 σ), coinciding with enhanced leaf fall (+1.4 σ), which were only locally apparent in 2005. Regions in the Amazon basin that flushed leaves at the onset of both droughts (+0.9 σ ~ +2.0 σ), showed positive anomalies in remotely sensed enhanced vegetation index, while sun-induced fluorescence and vegetation optical depth were reduced. The previously observed counterintuitive response of canopy green-up during drought in the Amazon basin detected by many remote sensing analyses can therefore be a result of enhanced leaf flushing at the onset of a drought. The long-term estimates of leaf litterfall and stem growth point to a decline of stem growth and a simultaneous increase in leaf litterfall in the Amazon basin since 1982. These trends are associated with increased warming and drying of the Amazonian climate, and could point to a further decline in the Amazon carbon sink strength.

# 1 Introduction

Tropical forests, in particular in the Amazon basin, contribute substantially (~25%) to the terrestrial carbon sink (Brienen et al., 2015; Pan et al., 2011). The Amazon forest alone currently stores an estimated 100 to 115 Pg of carbon in living biomass and intact forests have taken up an additional net 0.43 Pg of carbon each year through tree stem growth since the 1980's (Feldpausch et al., 2012; Phillips et al., 2017). It thereby acts to reduce the impact of deforestation and fossil fuel emissions on the atmospheric $CO_2$ growth rate and mitigates global climate change (Phillips et al., 2017). Most land surface models

project that the Amazon carbon sink will be sustained throughout the 21$^{st}$ century, mainly driven by the positive effect of elevated atmospheric $CO_2$ on plant growth (i.e. $CO_2$ fertilization) (Holm et al., 2020; Rammig et al., 2010). Also forest plot inventory data suggests a persistent carbon sink in intact Amazonian forests (Phillips et al., 2008) although the sink strength (i.e. the rate of net carbon uptake) has been declining since the start of the 21$^{st}$ century (Brienen et al., 2015; Hubau et al., 2020). The decline of the carbon sink strength is mainly driven by increased tree mortality while tree growth remained

relatively stable (Brienen et al., 2015). This suggests that the positive effect of elevated atmospheric $CO_2$ on plant photosynthesis and growth may increasingly be cancelled out by other limiting factors, such as nutrient availability (Fleischer et al., 2019; Hofhansl et al., 2016; Lapola et al., 2009). Additionally, the Amazon region is experiencing a change in the hydrological cycle with increasing wet season precipitation and flooding, a decline of dry season precipitation, more frequent episodic droughts and increasing regional air temperatures (Cox et al., 2008; Fu et al., 2013; Gloor et al., 2013; Janssen et al.,

2020; Jiménez-Muñoz et al., 2016). In light of these observed changes in regional climate and forest functioning, it is highly uncertain whether intact Amazonian forest will continue to act as a carbon sink in the future or will become a net source of $CO_2$ that could amplify global climate change (Boisier et al., 2015; Fu et al., 2013; Janssen et al., 2020; Malhi et al., 2009b; Marengo et al., 2010).

## 1.1 How sensitive is tree growth to drought in Amazonian forests?

Past responses of the Amazon forest productivity to droughts have been studied using satellite remote sensing analyses and field observations but sometimes with conflicting results. For example, many field observations show clear reductions in tree stem growth during drought (Feldpausch et al., 2016; Hofhansl et al., 2014; Rifai et al., 2018) while others found no reductions in stem growth during a drought (Doughty et al., 2015a; Phillips et al., 2009). Remote sensing studies complemented field observations and provided useful insights into the responses of forest productivity and aboveground biomass to drought over

time on regional and global scales (e.g. Liu et al., 2018b; Saleska et al., 2007). However, as remote sensing techniques measure electromagnetic radiation, it is notoriously difficult to interpret an observed drought response in remote sensing data and translate this response into a quantifiable change in growth or ecosystem carbon uptake (Mitchard et al., 2009a, 2009b). Furthermore, different remote sensing sensors sometimes point to contrasting responses of forest productivity to drought and seem to be deviating from ground observations (Anderson et al., 2010).


The discrepancy between drought responses observed in remote sensing products can partly be explained by the electromagnetic spectrum that the sensors utilize, so that the retrieved signal is sensitive to different vegetation properties. Vegetation indices derived from multispectral sensors that utilize red and near-infrared bands in the spectrum are sensitive to vegetation greenness and consistently show canopy green-up during and just after drought (Gonçalves et al., 2020; Lee et al., 2013; Saleska et al., 2007; Yang et al., 2018). However, the apparent green-up during drought has been coined an artefact and has been attributed to changes in atmospheric properties during drought (Asner and Alencar, 2010; Samanta et al., 2010), to changes in sun-sensor geometry (Morton et al., 2014), and to structural changes in the forest canopy (Anderson et al., 2010). Furthermore, other evidence from remote sensing analyses also seem to contradict the Amazon green-up during drought hypothesis (Anderson et al., 2018; Xu et al., 2011). Firstly, sun-induced fluorescence (SIF), measured with hyperspectral sensors and regarded a good proxy of canopy photosynthesis, is generally found to decrease during drought (Koren et al., 2018; Lee et al., 2013; Yang et al., 2018). Secondly, remotely sensed passive and active microwave data show clear negative anomalies in vegetation optical depth (VOD) and radar backscatter in response to drought in the Amazon basin, both metrics are considered sensitive to vegetation water content and biomass (Frolking et al., 2011, 2017; Liu et al., 2018b; Saatchi et al., 2013). For example, monthly observations of remotely sensed radar backscatter showed clear negative anomalies during the 2015 drought in the central Amazon that were correlated to *in situ* observed declines of stem diameter growth (van Emmerik et al., 2017). There is currently a lack of understanding of how observed remote sensing responses to drought translate into actual responses of aboveground forest growth and functioning in tropical forests.

### 1.2 What is known about the drivers of stem and canopy growth?

Total plant growth or biomass production is commonly divided into leaf growth, stem and branch growth, fine and coarse root growth, as well as reproductive growth. Next to quantifying total biomass production, it is relevant to know how biomass production is partitioned, because biomass in short-lived leaves and fine roots has a much shorter residence time compared to biomass in stems, branches and coarse roots. In Neotropical forests, the relative allocation of carbohydrates to biomass production in the canopy, stem and roots varies both spatially with climate and differences in soil properties (Hofhansl et al., 2015, 2020) as well as over time with changes in water availability, air temperature and insolation (Doughty et al., 2014, 2015a; Girardin et al., 2016). Stem growth is mostly estimated using a combination of dendrometer measurements and allometric equations (e.g. Malhi *et al.*, 2009b). Canopy growth is often determined by quantifying the amount of litterfall that is collected in so-called litter traps (e.g. Chave *et al.*, 2010). In Neotropical forest plots, stem growth increases with soil phosphorus availability, soil clay fraction and mean annual precipitation (Aragão et al., 2009; Banin et al., 2014; Hofhansl et al., 2015; Quesada et al., 2009; Soong et al., 2020). In contrast, the spatial variability in canopy production between sites is not explained by differences in mean annual precipitation or soil properties (Chave et al., 2009). Therefore, the drivers of the spatial variability in canopy growth across Neotropical forests remain largely unknown.

In humid Amazonian and other Neotropical forests, leaf flushing in the early dry season results in the increase of canopy growth and a simultaneous decline in stem growth (Doughty et al., 2014; Girardin et al., 2016; Hofhansl et al., 2014). The decline of stem growth during the dry season in humid forests is not related to a decline in overall biomass production but is related to a shift in carbohydrate allocation from the root and stem towards the canopy (Doughty et al., 2014, 2015b). In tropical dry forests, leaf litterfall increases in the dry season and leaf flushing is delayed until the start of the wet season when soil water is replenished (Sanches et al., 2008; Selva et al., 2007). Furthermore, the rate of dry season litterfall is observed to be higher near to the forest edge compared to the interior, associated with drier and warmer microclimatic conditions near the forest edge (Schessl et al., 2008; Sizer et al., 2000). On more wind exposed sites in the neotropics, not seasonality but the sporadic occurrence of tropical storms is driving the temporal variability in litterfall and canopy growth (Heineman et al., 2015; Liu et al., 2018a; Veneklaas, 1991). Finally, hot and dry conditions associated with the warm phase (El Niño) of the El Niño Southern Oscillation (ENSO) and tropical North Atlantic sea surface temperature anomalies (Marengo et al., 2011) have been linked to periods of elevated litterfall (Detto et al., 2018; Thomas, 1999) and reduced stem growth in Neotropical forests (Feldpausch et al., 2016; Rifai et al., 2018; Vasconcelos et al., 2012). However, it is still uncertain whether drought-induced changes in biomass production that were observed in inventory plots across the Amazon basin, occurred on a larger regional scale in forests across the entire basin.

The aims of this study are to develop a novel dataset of stem growth and leaf litterfall observations across the Amazon forest and other Neotropical ecosystems and examine how leaf litterfall, leaf flushing and stem growth change in response to drought in the Amazon forest. Furthermore, we aim to reconcile *in situ* measurements of leaf litterfall, leaf flushing and stem growth with remote sensing data, and use an empirical model to estimate the impact of historical droughts and long-term climate trends in the Amazon basin on aboveground biomass production.

## 2. Methods

### 2.1 Inventory data

We searched the available literature using the Google, Google Dataset Search and Google Scholar search engines for reported stem growth and litterfall data collected between 1981 and 2019 at sites across tropical and sub-tropical South and Central America between 30º south and 30 º north. The search timespan was chosen to match that of the *ERA5-Land* climate dataset that provided the explanatory variables in the empirical models (see section 2.4). Search terms included: leaf litterfall, litterfall, litterfall production, stem growth, diameter growth, tree growth. Also, the Spanish and Portuguese literature was searched for studies that reported litterfall production with the key words: producción de hojarasca and produção de serapilheira, respectively.

Monthly values of stem growth and litterfall were extracted from existing datasets as well as published manuscripts and compiled into a new dataset together with the month and year of observation, site name, location and data source (see Data availability). The majority of monthly data was extracted from published figures in individual manuscripts using a publicly available digitizing tool (Rohatgi, 2018). When the measurement time spanned multiple months or years, for example tree census data (e.g. Brienen *et al.*, 2015), instead of a well-defined year and month of observation, we included the start and end date of the census interval in the dataset. Total fine litterfall (including leaves, fruits, flowers and twigs) and leaf litterfall were, whenever possible, separately retrieved from the literature. When only leaf litterfall or total fine litterfall was provided in the original study, which was the case for 123 out of 211 studies that reported litterfall data, the missing litterfall data was estimated from a linear relationship between leaf litterfall and total fine litterfall ($R^2 = 0.93$, $p < 0.01$, $n = 3034$, Figure S1). All litterfall and stem growth data was converted to Mg C ha$^{-1}$ month$^{-1}$ using 50% carbon content per unit of biomass. The database included 7228 individual observations of litterfall and 2732 observations of stem growth that were retrieved from 246 studies conducted at 814 sites in the neotropics.

Litterfall observations can be used to estimate canopy growth at a specific site on multi-year timescales. However, monthly litterfall cannot be directly used to estimate monthly canopy growth as shed leaves are not instantly replaced by the same amount of newly flushed leaves. Therefore, we estimated monthly leaf flushing or leaf growth following Doughty & Goulden (2009) as:

$$Leaf flush = \frac{\Delta LAI}{SLA} + Leaf\ litterfall \qquad (1)$$

where $Leaf\ litterfall$ is the measured leaf litterfall (Mg C ha$^{-1}$ month$^{-1}$), SLA is the specific leaf area (m$^2$ Mg$^{-1}$ C) and $\Delta LAI$ the monthly change in leaf area index (m$^2$ ha$^{-1}$ month$^{-1}$). Specific leaf area data was extracted from the global gridded plant traits product of Butler et al. (2017). Monthly LAI was extracted for each site from July 1981 until December 2018 from the Global Data Set of Vegetation Leaf Area Index (LAI3g) (Zhu et al., 2013). The LAI3g is a validated global product developed using multi-spectral remote sensing data in a neural network algorithm, showing reasonable accuracy (RMSE = 0.68 m$^2$ m$^{-2}$) at ground truthing sites in various biomes and no saturation of LAI in dense broadleaf tropical forests (Zhu et al., 2013). In addition to leaf flushing, we estimated the proportion of mature leaf area as:

$$LAI_{mature} = \sum_{n=-5}^{-2} (Leaf flush * SLA)_n \qquad (2)$$

In Neotropical humid forests, newly flushed leaves take approximately two months to fully mature and reach their optimal photosynthetic capacity about 2-5 months after leaf flushing (Albert et al., 2018). Therefore, the sum of leaf area flushed

between 2 and 5 months in the past, here termed the mature leaf area, was thought to be a proxy of canopy photosynthetic capacity and canopy greenness.

## 2.2 Geospatial data and derived features

Properties that were not observed at the field plots included in the dataset (see section 2.1) were extracted from multiple gridded
geospatial datasets, including soil properties, plant traits standing biomass and climate data (Table 1). We included a broad range of geospatial variables that could possibly be used to predict the spatial and temporal variability in stem growth and leaf litterfall. However, the remote sensing products that were used in the comparison with the model output, the MODIS EVI, the vegetation optical depth and sun-induced fluorescence (see section 2.3), were not used as explanatory variables in the model to prevent interdependencies to occur between the model output and the remote sensing data.

Climate variables were retrieved as monthly averages from January 1981 to September 2019 at a 0.1° horizontal resolution from the *ERA5-Land* reanalysis dataset (Hersbach et al., 2020). In addition, hourly averages of instantaneous 10-meter wind gust were retrieved from January 1979 to September 2019 at a 0.25° horizontal resolution from the *ERA5* dataset. From the hourly averages of wind gust, the maximum wind gust in each month was calculated, which is expected to be a good indicator
of sporadic high litterfall following tropical cyclones (e.g. Whigham *et al.*, 1991).

**Table 1 Geospatial datasets used as explanatory variables in the XGBoost models. In brackets the native horizontal resolution of the dataset if a spatially aggregated product was used. The SoilGrids dataset (Hengl et al., 2017) contains data from seven soil layers at different depths below the surface. For this study, these layers were merged into two layers with a shallow soil layer (L1-L3) and a**
**deep soil layer (L4-L7).**

| Product name | Variables | Horizontal resolution | Temporal coverage | Data source |
|---|---|---|---|---|
| Plant traits | Specific leaf area ($m^2 kg^{-1}$)<br>Leaf nitrogen (mg $g^{-1}$)<br>Leaf phosphorous (mg $g^{-1}$) | 0.5° ~56 km | - | (Butler et al., 2017) |
| ESA CCI Aboveground biomass | Aboveground biomass (Mg $ha^{-1}$) | 500 m (100 m) | 2017 | ESA Climate Change Initiative<br>(Santoro and Cartus, 2019) |
| NASA Vegetation Continuous Field v1 (VCF5KYR) | Percentage tree cover (%) | 0.05° ~5.6 km | 1982-2016 | (Hansen and Song, 2018) |
| ALOS elevation and terrain | Elevation (m above sea level)<br>Slope (°)<br>Aspect (°) | 1 km (90 m) | 2006-2011 | (Tadono et al., 2014) |
| SoilGrids - global gridded soil data, second version (2017) | Available soil water capacity (%)<br>Cation exchange capacity (cmol $kg^{-1}$)<br>Bedrock depth (cm)<br>Clay, sand and silt fractions (%) | 1 km (250 m) | - | (Hengl et al., 2015, 2017) |

| | pH measured in water (index)<br>Organic carbon content (g kg$^{-1}$)<br>Total nitrogen (g kg$^{-1}$) | | | |
|---|---|---|---|---|
| GFPLAIN250m | Floodplain presence | 250 m | - | (Nardi et al., 2019) |
| ERA5 hourly averaged data from 1979 to present | Instantaneous 10 meter wind gust (m s$^{-1}$) | 0.25° ~28 km | 01-01-1979<br>01-09-2019 | (Hersbach et al., 2020) |
| ERA5-Land monthly averaged data from 1981 to present | 10 meter windspeed (m s$^{-1}$)*<br>Dewpoint temperature at 2m (K)<br>Temperature at 2m (K)<br>Evaporation (m of water equivalent)<br>Leaf area index high vegetation (m$^2$ m$^{-2}$)<br>Surface latent heat flux (J m$^{-2}$)<br>Surface net solar radiation (J m$^{-2}$)<br>Surface sensible heat flux (J m$^{-2}$)<br>Total precipitation (m)<br>Volumetric soil water in four layers (m$^3$ m$^{-3}$)<br>Skin reservoir content (m) | 0.1° ~11 km | 01-01-1981<br>01-09-2019 | (Hersbach et al., 2020) |

The number of explanatory variables, from here on called features, was further expanded by calculating derived features from the aforementioned datasets. Providing the empirical model with a large variety of often related features helps building performant models with a relatively low number of dependent variables (Guyon and Elisseeff, 2003). The soil C:N ratio was

calculated by dividing soil organic carbon content (g kg$^{-1}$) by soil total nitrogen content (g kg$^{-1}$) from the SoilGrids dataset (Hengl et al., 2017). Furthermore, the leaf N:P ratio was calculated from the leaf nitrogen (mg g$^{-1}$) and leaf phosphorus content (mg g$^{-1}$) present in the global gridded plant trait dataset (Butler et al., 2017). The gridded leaf N:P ratio was included into the empirical model as the gradient in plant available phosphorus is a key driver of forest structure and productivity across the Amazon basin (Quesada et al., 2012). Finally, the distance to the forest edge was calculated from the 500-meter horizontal

resolution aboveground biomass map as the Euclidean distance between every cell and the nearest cell with an aboveground biomass value below an arbitrary threshold of 50 Mg biomass ha$^{-1}$ (considered not forest). Because of the relatively high horizontal resolution (500 m) of the aboveground biomass map, the distance to the forest edge could not only identify the distance to large clearings and transitions to more open biomes but also the distance to smaller clearings and rivers.

To further expand the number of features available to train the model and to include historical climate data to the model, all monthly climate data up to 1 year in the past were separately added to the model. In this way, the model cannot only choose to use, for example, total precipitation in the present month but also the total precipitation in the previous month and the total presentation in the same month one year in the past to model stem growth and leaf litterfall in that particular month.

## 2.3 Remote sensing data

Reconciling differences between remote sensing observations from different sensors, as well as reconciling field and remote sensing observations required long-term records of remote sensing products from different sensors. The enhanced vegetation index (EVI) from the moderate resolution imaging spectroradiometer (MODIS) vegetation index product (MOD13C2 version 6) was used as an indicator of vegetation greenness (Gao et al., 2000). EVI is regarded as an improved vegetation index compared to the normalized difference vegetation index (NDVI), as it relies on the blue band next to the red and near infrared

bands and uses aerosol resistance coefficients in its formulation (Huete et al., 2000). The data were acquired from the website of the United States Geological Survey on a 0.05° grid with a 16 day temporal resolution from February 2000 up to April 2020. The pixel reliability layer that comes with MOD13C2 product was used to mask out all EVI pixels with unreliable data, keeping only the most reliable data (pixel reliability = 0) (Didan, 2015). Hereafter, the images were averaged to monthly values to be able to compare the EVI to the empirically modelled stem growth, leaf litterfall and leaf flushing data.


In addition, we used remotely sensed sun-induced fluorescence (SIF) data as a proxy of canopy photosynthesis. The SIF data used was retrieved from the recent Sun-Induced Fluorescence of Terrestrial Ecosystems Retrieval version 2 dataset (SIFTER v2). The SIF measurements are derived from hyperspectral observations of the GOME-2 sensor onboard the Metop-A satellite (Schaik et al., 2020). Monthly point observations of SIF (January 2007 - December 2016) were projected on a 0.5° global grid

and spatially aggregated to monthly averages for comparison with the field data and other remote sensing datasets.

Finally, monthly data was also available for vegetation optical depth (VOD), a passive microwave product (Liu et al., 2013; Meesters et al., 2005). VOD is directly proportional to the vegetation water content, and therefore sensitive to canopy density and biomass (Jackson and Schmugge, 1991; Meesters et al., 2005; Owe et al., 2001). Furthermore, the advantage of also using

VOD compared to the MODIS EVI is that VOD is unaffected by cloud cover. VOD has been used to study vegetation phenology (Jones et al., 2011, 2014) and to monitor global vegetation dynamics (Andela et al., 2013; Liu et al., 2007, 2013, 2015) and deforestation (van Marle et al., 2016). We used C band (June 2002 – December 2018) and X band (December 1997 – December 2018) VOD data from the global long-term Vegetation Optical Depth Climate Archive (Moesinger et al., 2020).

## 2.4 Data analysis

Machine learning enables integrating the different spatial and temporal scales inherent to the field observations in a single method and making predictions based on the trends identified in the data. Extreme gradient boosting (XGBoost), a machine learning method for classification and regression (Chen and Guestrin, 2016) was used to upscale *in situ* measurements to estimate monthly leaf litterfall and stem growth rates for the neotropics from 1982 to 2019.

The XGBoost algorithm was selected for its demonstrated performance when applied to similar environmental science problems such as soil mapping (Hengl et al., 2017) and estimating evapotranspiration (Fan et al., 2018). Like other boosting algorithms, XGBoost uses an ensemble of weak prediction models, iteratively building each new model to improve the prediction of the ensemble of previous models. In essence, XGBoost constructs a series of relatively shallow regression trees that provide a continuous output value at each leaf, these output values are summed over all regression trees to derive the final prediction. The output value of each regression tree is scaled by a predetermined factor η (learning rate) which reduces the weight of the individual tree. Adjusting this factor vigilantly ensures a smooth descent of the loss function (Chen and Guestrin, 2016). Besides the learning rate, XGBoost enables the use of multiple other regularization options. The parameters modulating the regularization options in the model (so-called hyperparameters) are tuned to make the final model more robust and prevent overfitting on the training data. Here, we use the R package *xgboost* (Chen et al., 2020) to construct the model and the R package *mlr* (Bischl et al., 2020) to tune hyperparameters and select the final features used in the model.

Two XGBoost models were constructed, to estimate monthly leaf litterfall and stem growth separately. Before setting up the models, the stochastic behaviour present in the monthly timeseries of leaf litterfall and stem growth was reduced by using a moving average filter with a window size of 3 months. The 3-month window size, the lowest possible window size, was chosen to reduce the sometimes large month to month variation in leaf litterfall and stem growth while maintaining sufficient variation between consecutive months to identify extremes. Furthermore, positive outliers, defined as values higher than 3 times the standard deviation above the mean, were omitted. The monthly climate data linked to the stem growth and leaf litterfall observations spanning multiple months to years was averaged using the start date and end date of the observation interval. To account for the difference in observation timespan, weights were assigned to the observations in the model as following:

$$Observation weight = 1 + ln(n_{months}) \qquad (3)$$

where $n_{months}$ is the length of the time interval in months. By using the natural logarithm to assign weights, observations covering multiple months to years were assigned 2 to 5 times the weight of a monthly observation. This was preferred in contrast to assigning weights directly proportional to the length of the time interval as this would inflate the importance of a few sites with very long observation time intervals in the model.

Model performance was evaluated by dividing all leaf litterfall and stem growth data into a training dataset containing 60% of all observations at each site and a test dataset containing the remaining 40% of the observations. The initial XGBoost model was constructed using the default learning rate (0.3) and the best model iteration was estimated using a 10-fold cross-validation of the training data, selecting the iteration with the lowest root mean squared error (RMSE) on the cross-validated data. Next, we filtered out 80% of the initial 235 features with the lowest feature importance (gain) to reduce the dimensionality of the data and speedup subsequent tuning. Hyperparameter tuning of all the model parameters was done by random search using

1000 iterations and 10-fold cross-validation. Subsequently, feature selection was done to select a maximum of 20 features for each model with the updated hyperparameters and random search using 1000 iterations and 10-fold cross-validation. Furthermore, to derive an estimate of model uncertainty, two additional XGBoost models were trained and similarly tuned to estimate the model error, which is defined as the squared difference between the observed value and the predicted value in the test dataset. The final models with the tuned hyperparameters and the 20 selected features were also trained on a separate training dataset containing data from 60% of the sites (instead of 60% of the data from each site) to validate model performance for between site variation. In this second validation procedure, complete time series of 60% of the sites were used as training data to estimate complete time series for 40% of the remaining sites (Figure S2).

To evaluate the drought responses of modelled stem growth, leaf litterfall and leaf flushing, two rectangular drought areas were delineated within the Amazon basin for the 2005 and the 2015 drought period. First, the drought period was identified for both droughts using the average ERA5 topsoil moisture content for the entire Amazon basin. For each month in the time series, the seasonally detrended topsoil moisture content was calculated by subtracting the monthly average and dividing by the standard deviation of that month. The drought period was defined as the consecutive months with a topsoil moisture content below one standard deviation ($\sigma$) compared to its monthly average. Subsequently, a rectangular area was delineated that overlapped those areas within the Amazon basin that showed a topsoil moisture content $< 1.5\ \sigma$ averaged over the entire drought period.

## 3. Results

### 3.1 Model evaluation and feature importance

The two XGBoost models, one for stem growth (NRSME = 10.6%) and one for leaf litterfall (NRMSE = 9.4%), showed a comparable accuracy across the 40% of the data that were used to evaluate the models (Figure 1a, 1c). The model predicting stem growth showed less uncertainty in absolute metrics (RMSE = 0.06 Mg C ha$^{-1}$ month$^{-1}$) compared to the model predicting leaf litterfall (RMSE = 0.08 Mg C ha$^{-1}$ month$^{-1}$). However, the range in observed values and the explained variation was smaller for the stem growth model ($R^2 = 0.54$) compared to the leaf litterfall model ($R^2 = 0.71$). The XGBoost models validated for estimating between site variation, in which the test data did not include the same sites as the training data, showed lower performance in estimating stem growth (RMSE = 0.06 Mg C ha$^{-1}$ month$^{-1}$, $R^2 = 0.41$, Figure S2 a) and especially leaf litterfall rates (RMSE = 0.12 Mg C ha$^{-1}$ month$^{-1}$, $R^2 = 0.4$, Figure S2 b). This additional model validation reveals that the two models perform better when trained on incomplete time series from all available sites compared to complete time series from a selection of sites. This in turn suggests that the drivers of the temporal variation in stem growth and especially leaf litterfall are well represented by the set of features used in the models while the drivers of the spatial variation are not fully included.

In both models, high rates of stem growth and leaf litterfall were consistently underestimated while relatively low values were overestimated (Figure 1a, 1c). This is a common problem in machine learning as the variance of the model estimates is always lower (unless the fit is perfect) compared to the variance of the observations resulting in the model estimates moving closer to the observed mean. The underestimation of high values and overestimation of low values of stem growth and leaf litterfall is a limitation of this method when using it to study extreme events like droughts, when extreme responses of stem growth and

leaf litterfall are expected. Therefore, the results presented here should be considered a conservative or lower bound estimate of the actual responses of leaf litterfall and stem growth to drought that are observed.

The number of observations of leaf litterfall and stem growth per year are not evenly distributed over time in the dataset (Figure S3 a). The frequency of leaf litterfall and stem growth measurements in the dataset increased in the 1980's and the 1990's to

a maximum in the first decade of the 21$^{st}$ century and has since steadily declined (Figure S3 a), presumably as a proportion of the more recent data has not yet been published or is still under embargo. Despite the increase in observation frequency over time, the model uncertainty, expressed as the NRMSE, has significantly increased over time since the 1980's, both for leaf litterfall (r = 0.6, p < 0.001) and stem growth (r = 0.4, p < 0.05, Figure S3 b). Suggesting that the model estimates of leaf litterfall and stem growth are relatively more uncertain in recent years compared to the 1980's and 1990's.


Of the 235 features that were used in the first XGBoost models, only 20 features were used in the two final models. These features have been ranked based on their importance (gain) in these final models and the top tens of most important features in both models are shown (Figure 1b, 1d). The most important features explaining the spatial and temporal variability in stem growth and leaf litterfall that were used in both models included terrain elevation, soil moisture content, vapour pressure

deficit, sensible heat flux, solar radiation, leaf nitrogen : phosphorus ratio, and percentage tree cover. Additional features explaining stem growth included precipitation and evaporation, terrain aspect, bedrock depth and soil pH (Figure 1b). The spatial and temporal variability in leaf litterfall was further explained by features including aboveground biomass, meteorological variables such as dewpoint and air temperature and wind speed and soil properties such as soil nitrogen content and soil clay fraction (Figure 1d). Although the importance of some of these features in the models might represent a causal

link with either stem growth or leaf litterfall, we cannot conclude from this empirical analysis that this is the case.

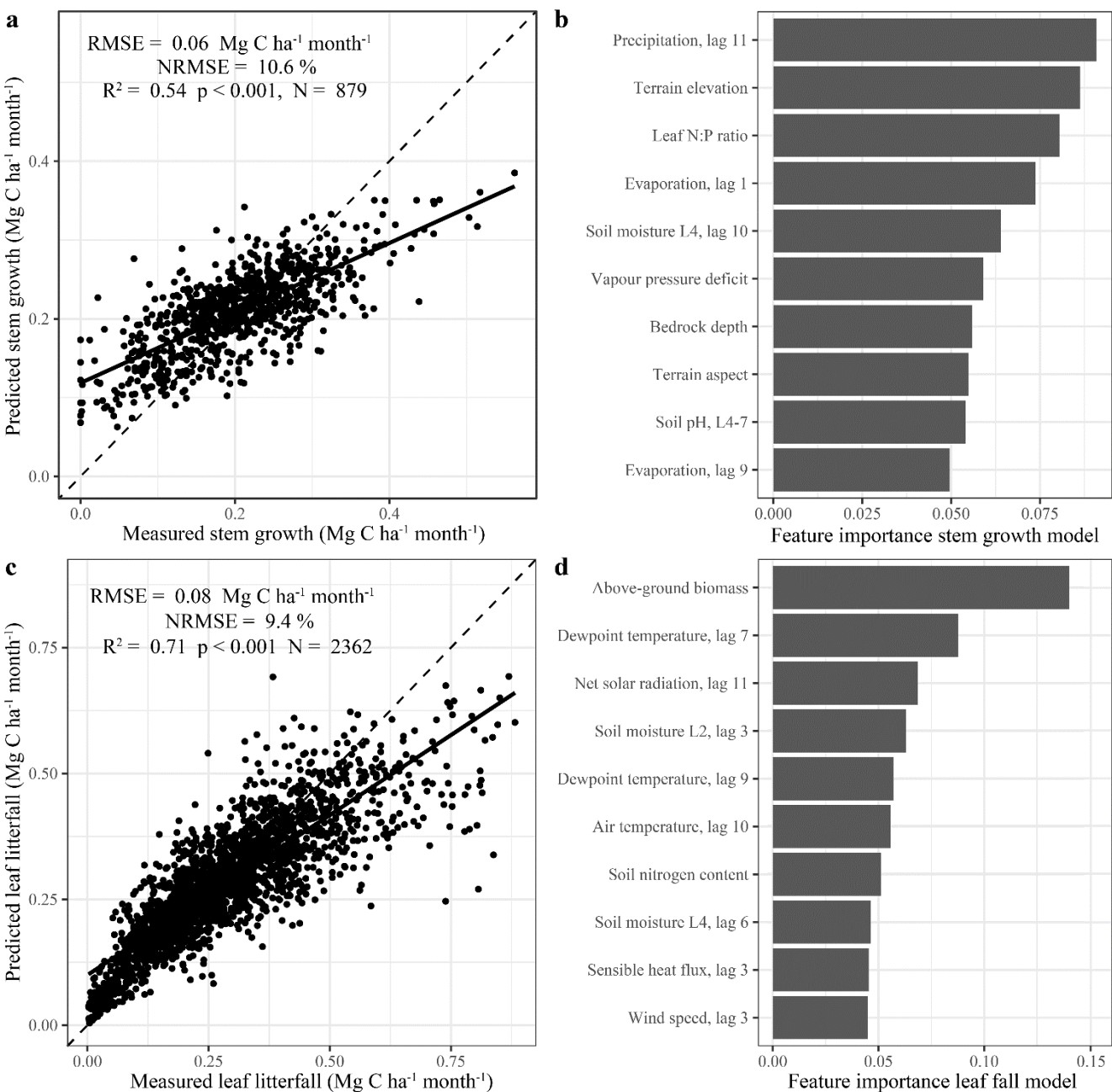

**Figure 1 Model evaluation and feature importance. The scatterplots on the left side of the figure (a, c) show the predicted biomass production versus the measured biomass production of the test data that was used to validate the stem growth (a) and leaf litterfall (b) models. The dashed black line is the 1:1 line and the solid black line the least squares linear regression fit. The bar graphs on the right side of the figure (b, d) show the feature importance (gain) of the top 10 features selected for the final models. Feature names are detailed in Table 1. Features with lags indicate the value of that climate variable a given number of months in the past (e.g. precipitation lag 11 is the monthly precipitation 11 months in the past).**


### 3.2 Long-term stem growth and leaf litterfall rates across the neotropics

Distinct spatial patterns in stem growth and leaf litterfall rates across the neotropics arose in the long-term (1982-2019) model estimates (Figure 2). The range of predicted leaf litterfall rates (0.8 ~ 5.9 Mg C ha$^{-1}$ year$^{-1}$) across the Neotropics was almost two times as large as the range of predicted stem growth (1.0 ~ 3.6 Mg C ha$^{-1}$ year$^{-1}$), in accordance with the observed difference in the range of the field data (Figure 1a, 1b). Although the spatial patterns in stem growth and litterfall rates differed, some general trends can be identified. Relatively low rates of predicted stem growth and leaf litterfall are observed in the open

savanna and xeric shrub ecosystems of the Neotropics such as the Cerrado and Caatinga in Brazil, the Llanos savanna in Venezuela and the Beni savanna in Bolivia (Figure 2a , 2b). Furthermore, low stem growth and leaf litterfall rates are also observed in the montane environments of the Andes (Figure 2a, 2b). Relatively high rates of predicted stem growth are found in Central America, along the Pacific coast of Colombia and in the northern and western Amazon basin (Figure 2a). Leaf litterfall showed relatively high rates in the remaining Atlantic forest fragments of south-eastern Brazil, in Central America

and across the forest covered Amazon basin (Figure 2b).

As the range in predicted leaf litterfall rates was much larger than the range in predicted stem growth rates, the spatial variability in leaf litterfall rates largely drives the spatial variability in aboveground biomass production (defined as the long-term sum of leaf litterfall and stem growth) across the Neotropical ecosystems (Figure 2c). Furthermore, the predicted stem growth and

leaf litterfall data shows that in areas with a relatively low aboveground biomass production, for example in the Cerrado region and the Andes, the contribution of stem growth to the total aboveground biomass production is relatively large (> 0.45). In contrast, in areas where aboveground biomass production is relatively high, for example in the Amazon basin and Central America, the contribution of stem growth to the total aboveground growth is relatively low (< 0.45, Figure 2d). These results suggest that as productivity increases in these Neotropical ecosystems, an increasingly larger proportion of available

carbohydrates is allocated to the production of leaves.

The estimated model uncertainty (RMSE) of the stem growth and leaf litterfall models showed similar spatial patterns as the long-term averages with a high RMSE in highly productive regions and low RMSE in less productive regions (Figure S4 a ,b). However, after adjusting the RMSE for the average seasonal range in values observed (the annual amplitude), it becomes clear

that the relative error (NRMSE) is actually higher in the unproductive regions, especially in the Andes (Figure S4 c, d). While the leaf litterfall and stem growth models show good performance in the majority of pixels in the study area (NRMSE < 15%), some high altitude areas within the Andes show a relatively low performance (NRMSE > 50%). For the Amazon basin (black contour) the average estimated NRMSE is 12.5% for the leaf litterfall model and 16.4% for the stem growth model. This means that on average, the error of the model estimates across the Amazon basin is less than 20% of the average seasonal variability

in leaf litterfall and stem growth.

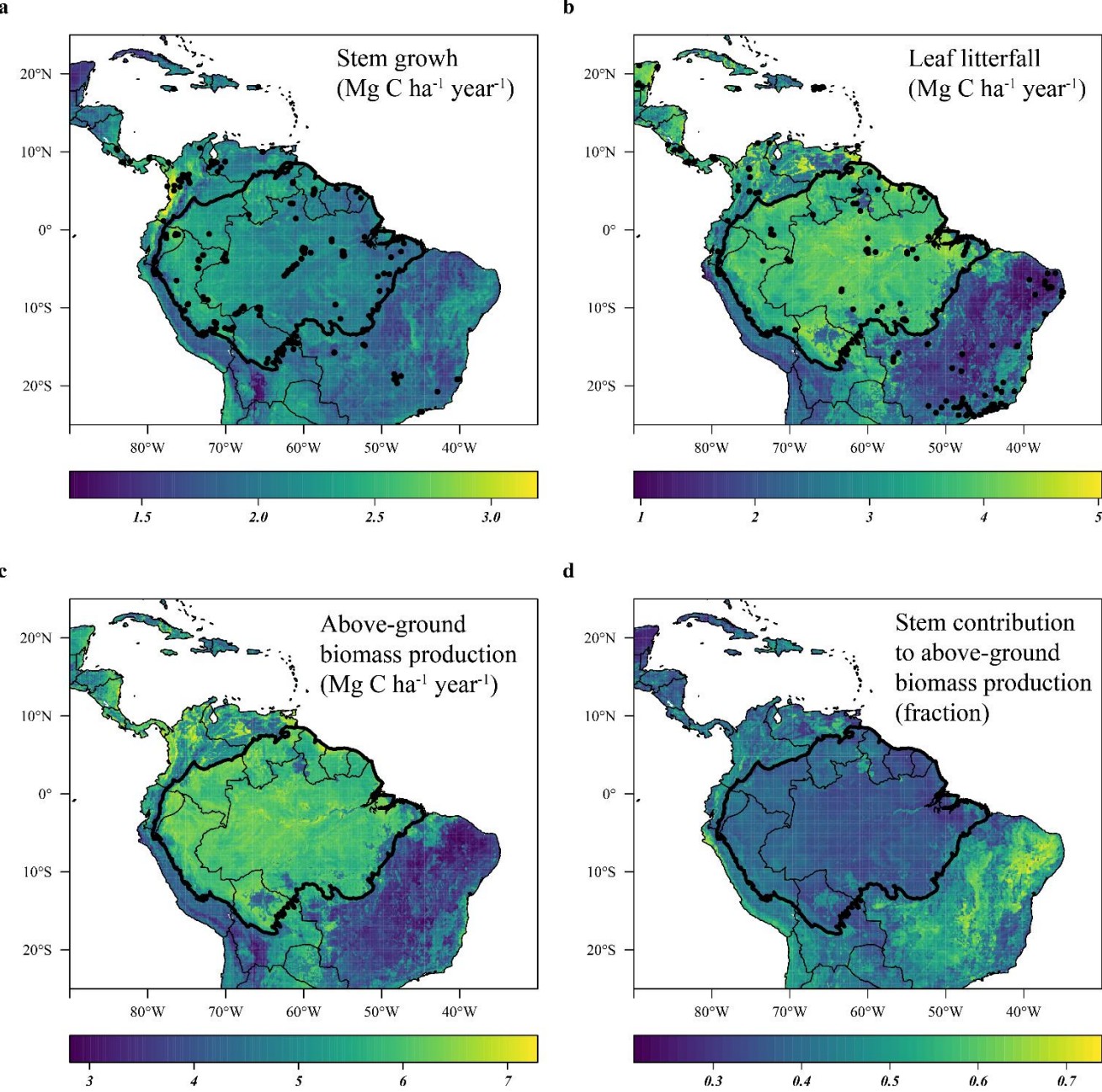

**Figure 2** Predicted stem growth (a), leaf litterfall (b) and total aboveground biomass production (c) and the contribution of stem growth to the aboveground biomass production (d) across the neotropics from 1982 to 2019. Site locations where stem growth (n = 458) (a) and leaf litterfall (n = 377) (b) were measured are depicted as solid black circles. Country borders and the extent of the Amazon basin are marked by thin and thick black lines, respectively.

## 3.3 Aboveground growth responses to the drought of 2015

The predicted monthly stem growth and leaf litterfall data were used to estimate the impact of the 2015 drought in the Amazon region. Across the entire Amazon basin, leaf fall generally showed positive anomalies while stem growth showed negative anomalies during the 2015 drought (August 2015 to January 2016, Figure 3a & 3c). However, significant regional differences in the responses of leaf fall, leaf flushing and stem growth to the 2015 drought were observed within the Amazon basin (Figure 3). A combination of positive seasonal anomalies in leaf fall and negative anomalies in stem growth during the 2015 drought were mainly observed in the eastern Amazon that was delineated as the drought area (red rectangle in Figure 3 and 4). This area experienced the most significant negative anomalies in top-soil volumetric moisture content and positive anomalies in net solar radiation (Figure 4a & 4b). During the height of the drought in November 2015, precipitation (-1.7 σ) and soil moisture (-2.6 σ) were significantly lower in the drought area compared to their monthly average, while air temperature (+2.7 σ), vapor pressure deficit (+2.8 σ) and solar radiation (+2.1 σ) were all significantly higher compared to their monthly average (Figure 5c).

From August 2015 to January 2016 stem growth was on average significantly lower (-1.8 σ) in the drought area while leaf fall was higher (+1.4 σ) compared to the long-term averages for these months (Figure 5a). In the dry season following the 2015 drought, from July 2016 to December 2016, stem growth was also significantly reduced in the area affected by drought (-2.1 σ) while leaf fall was again elevated (+1.7 σ) compared to their long-term averages. These results point to a lagged effect of the 2015 drought on leaf fall and stem growth. Leaf flushing was higher than the monthly average at the onset (+2.0 σ in August 2015) and end of the drought (+1.0 σ in January 2016) following the first rain events (Figure S5). During the height of the drought leaf flushing was lower than the monthly average (-1.1 σ in November 2015, Figure 5).

In the drought area, anomalously high leaf flushing at the onset of the 2015 drought resulted in an above average mature leaf area (i.e. the sum of leaf area flushed in the past 2-5 months), in the second half of the drought (+1.8 σ in September 2015 - January 2016, Figure 5a). The spatial pattern of the positive anomalies in mature leaf area coincided with positive anomalies in MODIS EVI (Figure 3d & 4c). Green-up during drought was visible as positive anomalies in predicted mature leaf area and EVI in eastern Colombia and in the central Brazilian Amazon, roughly the west half of the delineated drought area (Figure 3d & 4c). However, in the east half of the drought area, mainly negative anomalies in EVI, leaf flushing and mature leaf area were visible (Figure 3b, 3d & 4c). This area experiences a relatively long dry season (≥ 4 months) compared to the forest in the west (< 3 months) (Sombroek, 2001), suggesting that forests experiencing a short dry season green-up during drought while forests experiencing a longer dry season generally show browning in response to drought.

The X band vegetation optical depth (VOD) and sun-induced fluorescence (SIF) showed widespread negative anomalies in the drought area (-0.8 σ and -2.4 σ in September - November, respectively) during the height of the 2015 drought (Figure 4c

& 5b). Note the contrast in the observed responses between the moist tropical forest of the Amazon basin (inside the black contour line) with the Cerrado and Caatinga regions, located to the south and south-east of the Amazon basin in eastern Brazil. In the drier Cerrado and Caatinga regions, both VOD and EVI show clear negative anomalies during the 2015 drought (Figure 3 & 4).

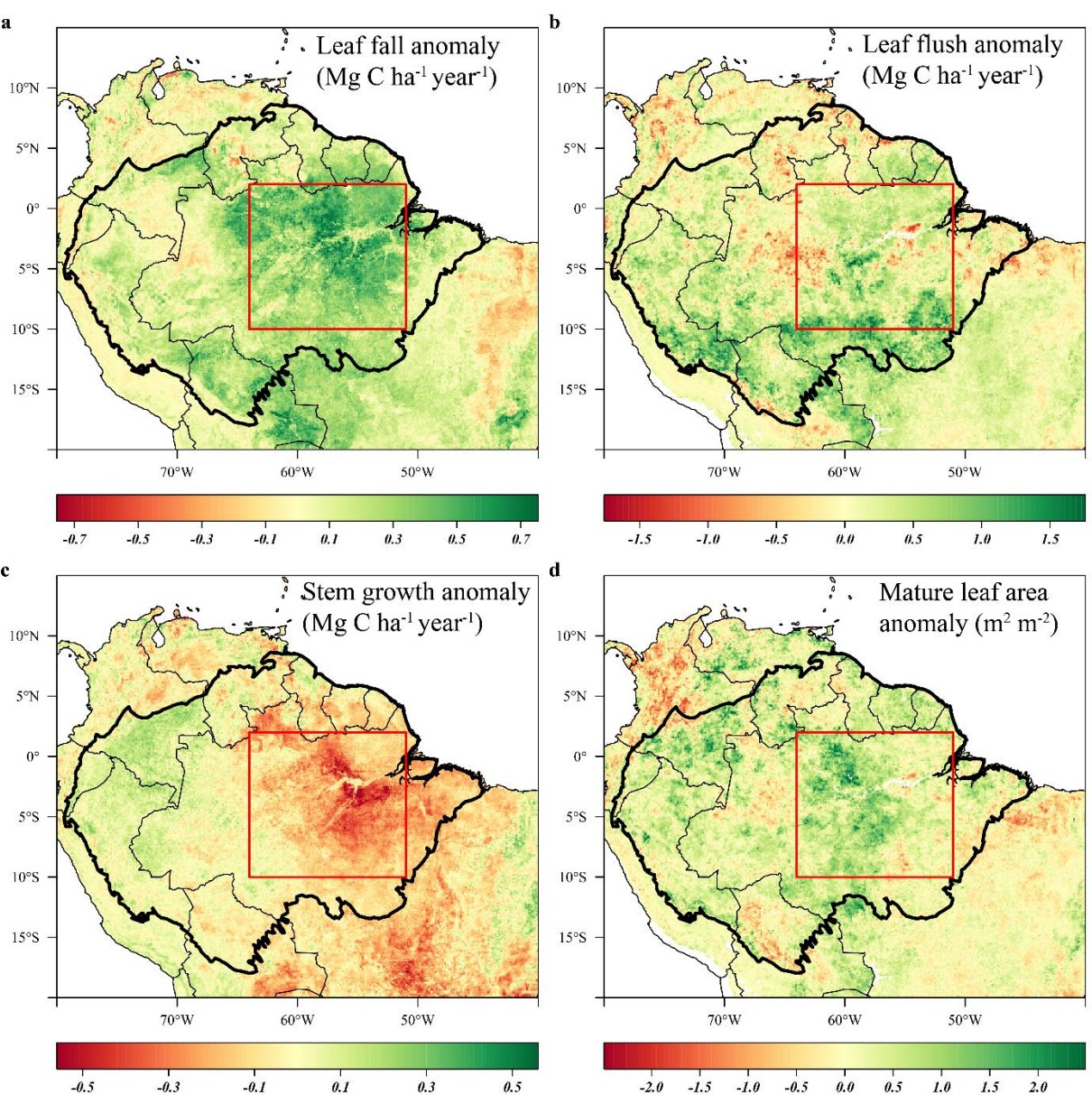

**Figure 3 Average anomalies in leaf fall, leaf flushing, stem growth and mature leaf area during the 2015 drought (August 2015 – January 2016) compared to their long-term averages (1982-2019). Leaf fall (a) and stem growth (c) were directly retrieved from the long-term monthly model estimates. Leaf flush (b) was calculated from monthly predicted leaf fall (a) and changes in LAI (Eq. 1). Mature leaf area (d) is the sum of new leaf area flushed in the previous 2 to 5 months (Eq. 2). Country borders and the extent of the**

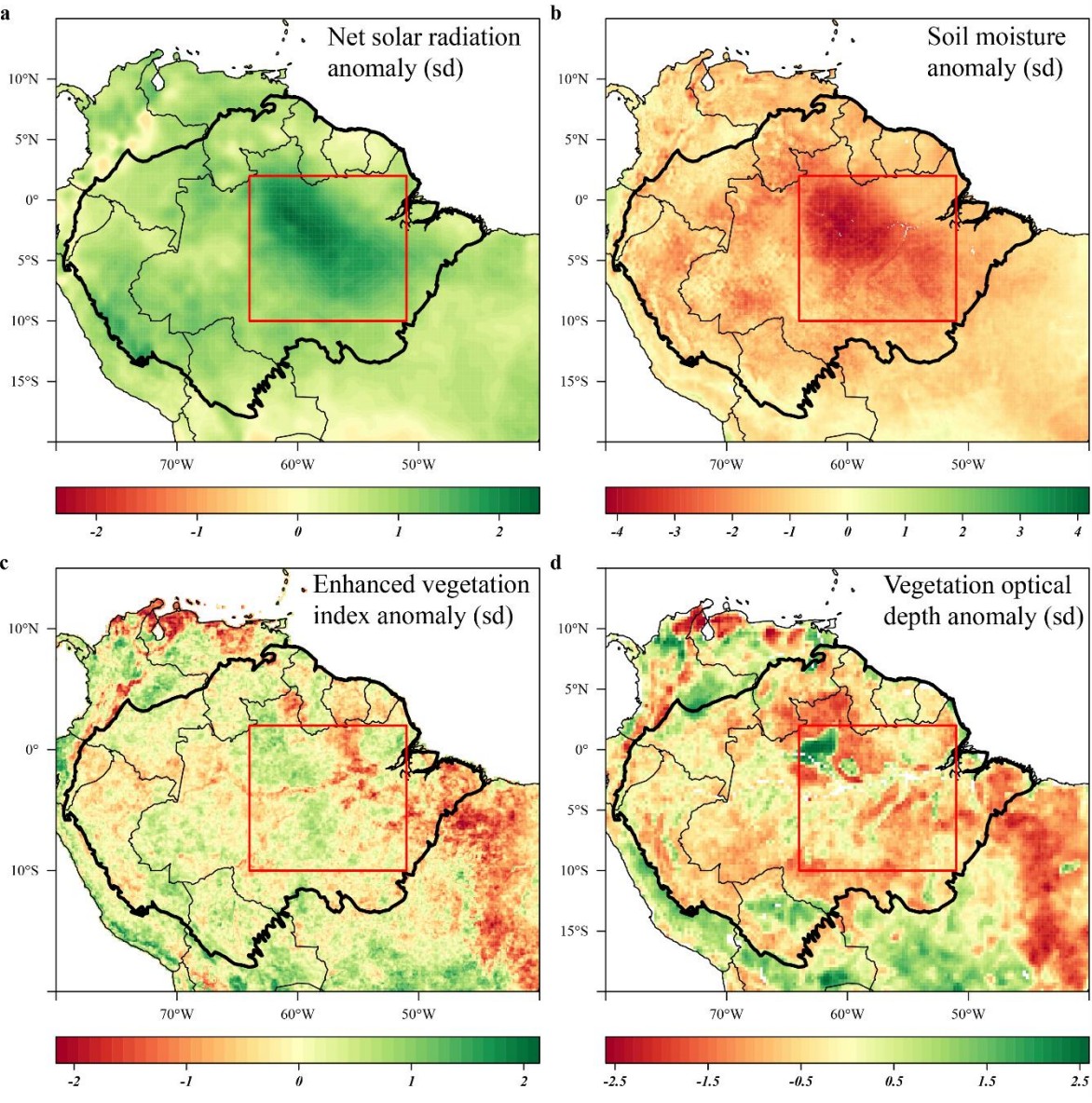

**Figure 4 Standardized anomalies in net solar radiation (a), soil moisture (b), enhanced vegetation index (c) and X band vegetation optical depth (d) during the 2015 drought (August 2015 – January 2016) compared to their long-term monthly averages. Soil moisture anomalies are calculated from the ERA-5 volumetric soil moisture in the first soil layer (L1). Country borders and the extent of the Amazon basin are marked by thin and thick black lines, respectively. The red rectangle delineates the drought area for which further results are reported.**

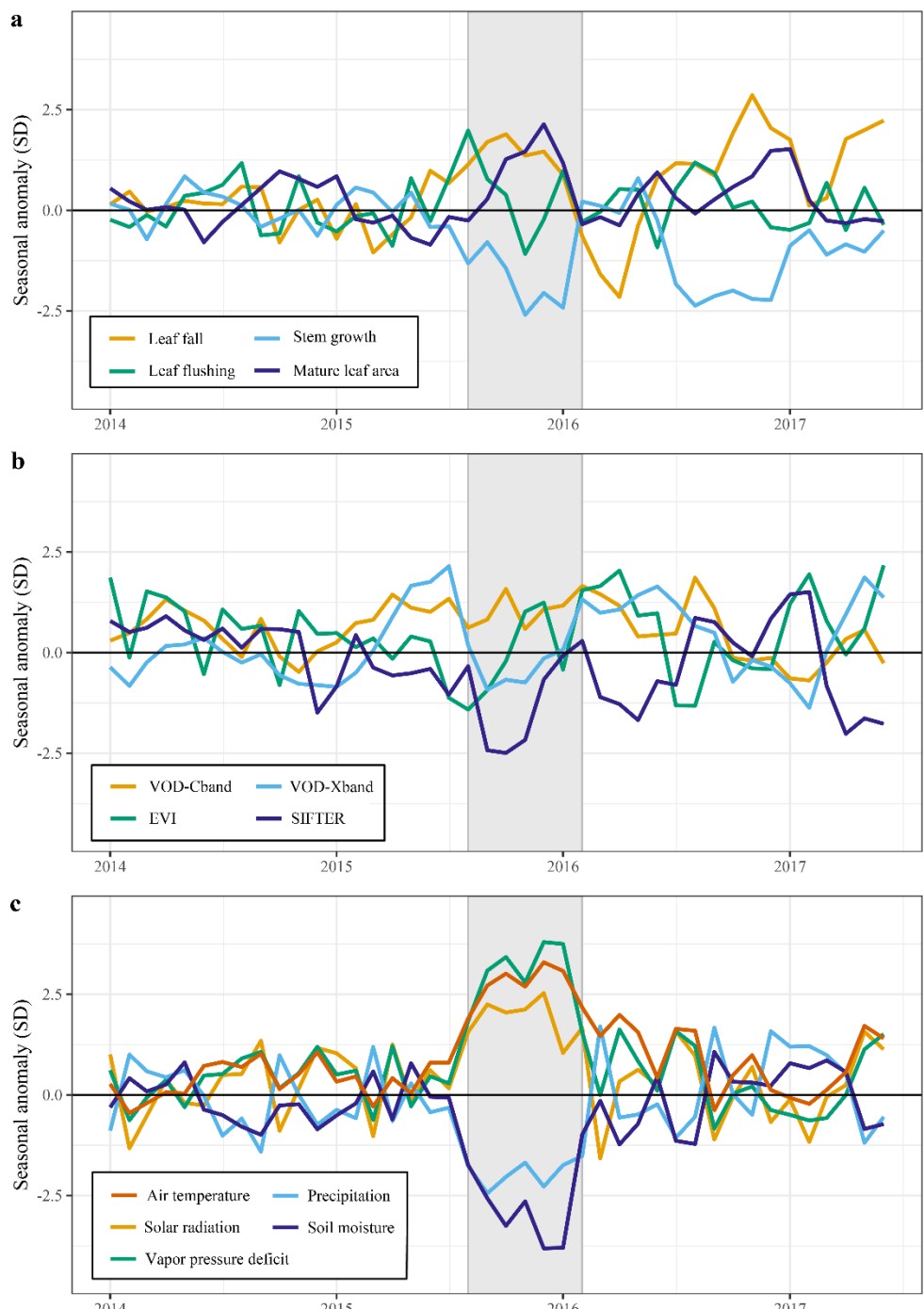

**Figure 5 responses of aboveground growth and remotely sensed vegetation properties to the 2015 El Niño drought and key climatic variables. All graphs show the trend in the standardized seasonal anomaly, the deviation from the monthly mean divided by the standard deviation of that month. Leaf fall and stem growth (a) are derived from the two separate XGBoost models providing monthly values from January 1982 until September 2019. Mature leaf area (a) is the sum of flushed leaves from 2 to 5 months in the past**

### 3.4 Aboveground growth responses to the drought of 2005

The long-term records of predicted leaf litterfall, leaf flushing and stem growth enable looking back at changes in estimated growth that occurred in response to other historic droughts. The drought of 2005 is considered a particularly severe drought in the western Amazon and was the first major drought captured by the MODIS sensors which led to the first observations of Amazon forest green-up during drought (Saleska et al., 2007).

Similar to the 2015 drought, the estimated leaf litterfall showed widespread positive anomalies in the 2005 drought area (Figure 6 a). However, in contrast to 2015, stem growth does not show consistent negative anomalies across the drought area (Figure 6 c). Leaf flushing shows mainly positive anomalies in the west of the drought area and negative anomalies in the east (Figure 6 b) while mature leaf area shows positive anomalies in south of the study area and negative anomalies in the north (Figure 6 d).

The new generation of algorithms and the longer time-series of MODIS EVI data confirm the findings of Saleska *et al.* (2007), i.e., that EVI was significantly and consistently higher during the 2005 drought, compared to the long-term average (Figure 7c). EVI was significantly elevated before and at the onset of the 2005 drought (+1.9 σ) in March to May 2005 and remained higher during the height of the 2005 drought (+1.3 σ) in June to August 2005 (Figure 8b). Similar to 2015, we find that X band VOD was significantly lower in the drought area during the height of the 2005 drought (-1.2 σ) in June to August 2005 while C band VOD did not show a clear effect of the 2005 drought (Figure 8b).

During the 2005 drought (June-September), precipitation (-1.5 σ) and soil moisture (-1.3 σ) were lower compared to their monthly averages in the drought area (Figure 7b & 8c). Air temperature (+0.9 σ), vapour pressure deficit (+1.4 σ) and solar radiation (+1.3 σ) were all higher during the 2005 drought compared to their monthly averages (Figure 7a & 8c). The duration of the 2005 drought (4 months) was shorter compared to the 2015-2016 drought (6 months) and when comparing the seasonal anomalies of the climatic variables in the drought areas, the 2015 drought was clearly more severe and more anomalous compared to the 2005 drought (Figure 5c & 8c). Approximately one year after the 2005 drought, another short drought hit this part of the Amazon basin, with significant negative anomalies in top-soil moisture content (-2.3 σ) and precipitation (-1.3 σ) and positive anomalies in VPD (+1.9 σ) between May and July 2006.

In the entire drought area, leaf flushing was higher at the onset (+0.9 σ in June 2005) and at the end of the drought (+0.8 σ in August - September  2005) and lower at the height of the drought (-1.3 σ in July 2005) compared to the long-term monthly average (Figure 8a). Enhanced leaf flushing at the onset of the 2005 drought resulted in a higher mature leaf area (+1.0 σ in August 2005) at the end of the drought (Figure 6b & 8a).

Compared to 2005, the above-ground growth responses were more pronounced during the short 2006 drought following the drought of 2005, with significant positive anomalies in leaf fall (+1.2 σ) and leaf flushing (+1.0 σ) and negative anomalies in stem growth (-1.2 σ) and X band VOD (-1.6 σ) in May to July 2006 (Figure 8a). Enhanced leaf flushing during and following this short 2006 drought resulted in higher than average mature leaf area (+1.5 σ) and EVI (+0.8 σ) in the months following the drought from August to November 2006 (Figure 8).

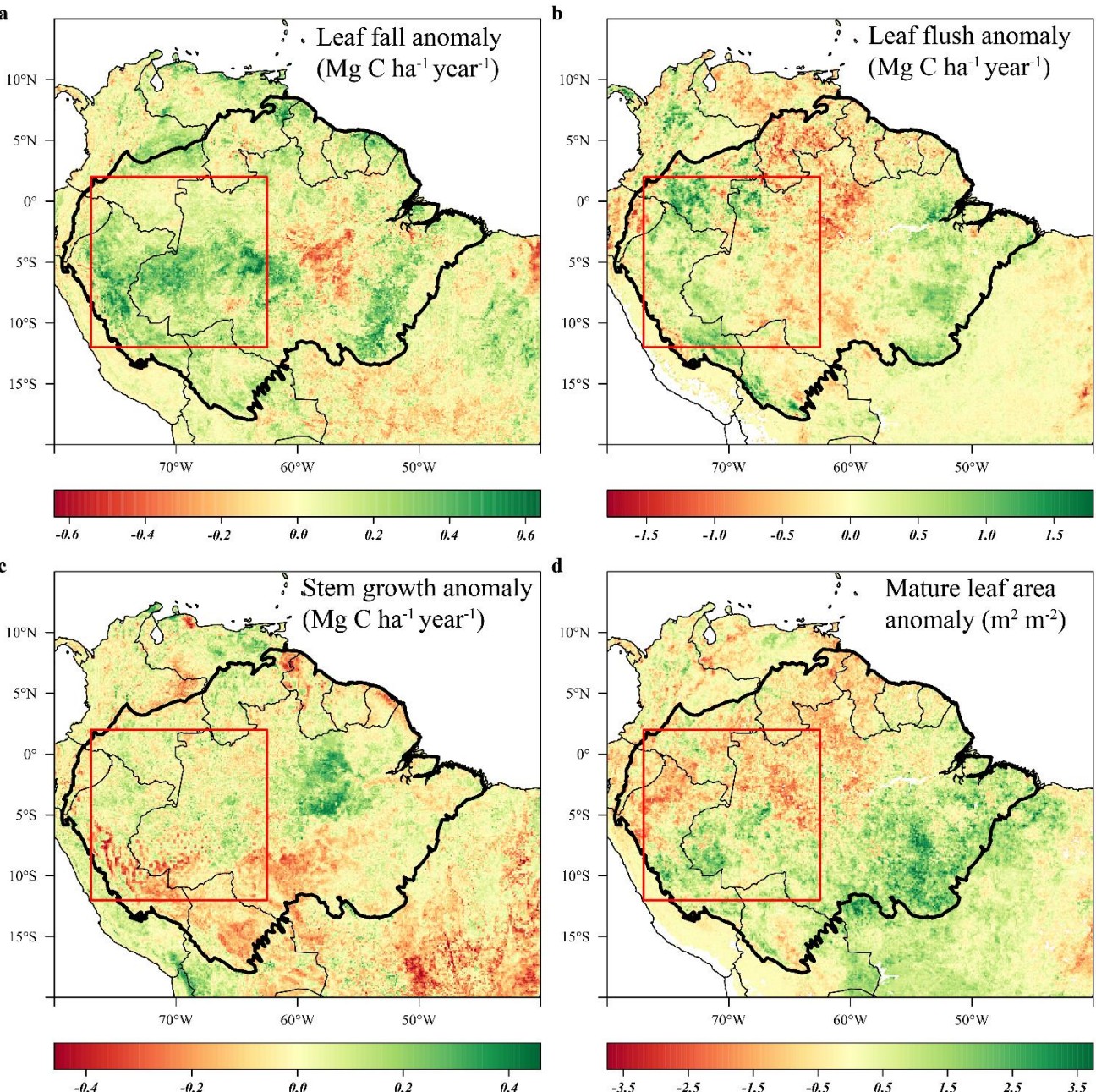

**Figure 6 Average anomalies in leaf fall, leaf flushing and stem growth and mature leaf area during the 2005 drought (June – September 2005) compared to their long-term monthly averages (1982-2019). Leaf fall (a) and stem growth (c) were directly retrieved from the long-term monthly model estimates. Leaf flush (b) was calculated from monthly predicted leaf fall (a) and changes in LAI. Mature leaf area (d) is the sum of new leaf area flushed in the previous 2 to 5 months. Country borders and the extent of the Amazon basin are marked by thin and thick black lines, respectively. The red rectangle delineates the drought area for which further results are reported.**

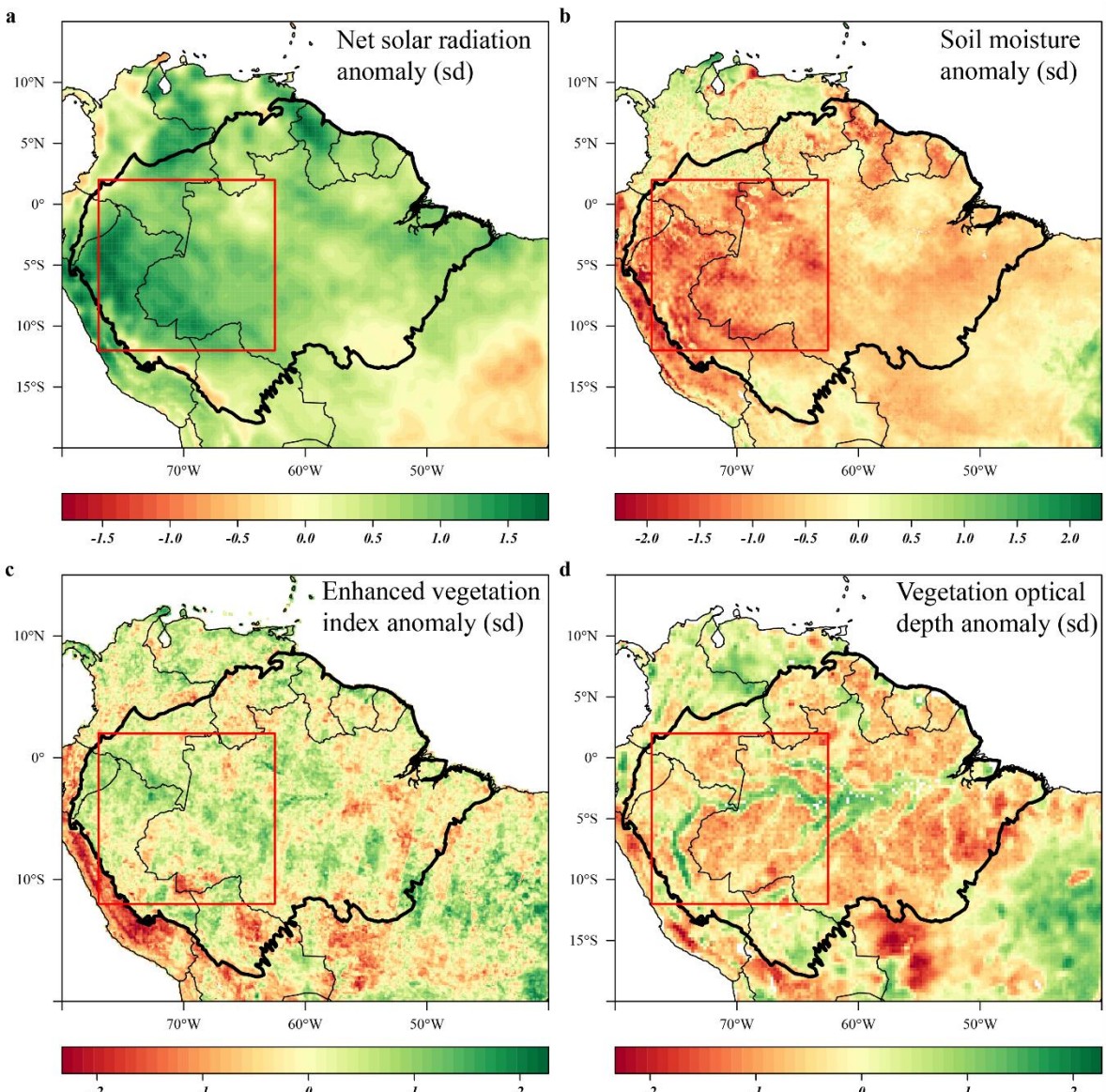

**Figure 7 Seasonal anomalies in net solar radiation (a), soil moisture (b), enhanced vegetation index (c) and X band vegetation optical depth (d) during the 2005 drought (June – September 2005) compared to their long-term monthly averages (1982-2019). Soil moisture anomalies are calculated from the ERA-5 volumetric soil moisture in the first soil layer (L1). Country borders and the extent of the Amazon basin are marked by thin and thick black lines, respectively. The red rectangle delineates the drought area for which further results are reported.**

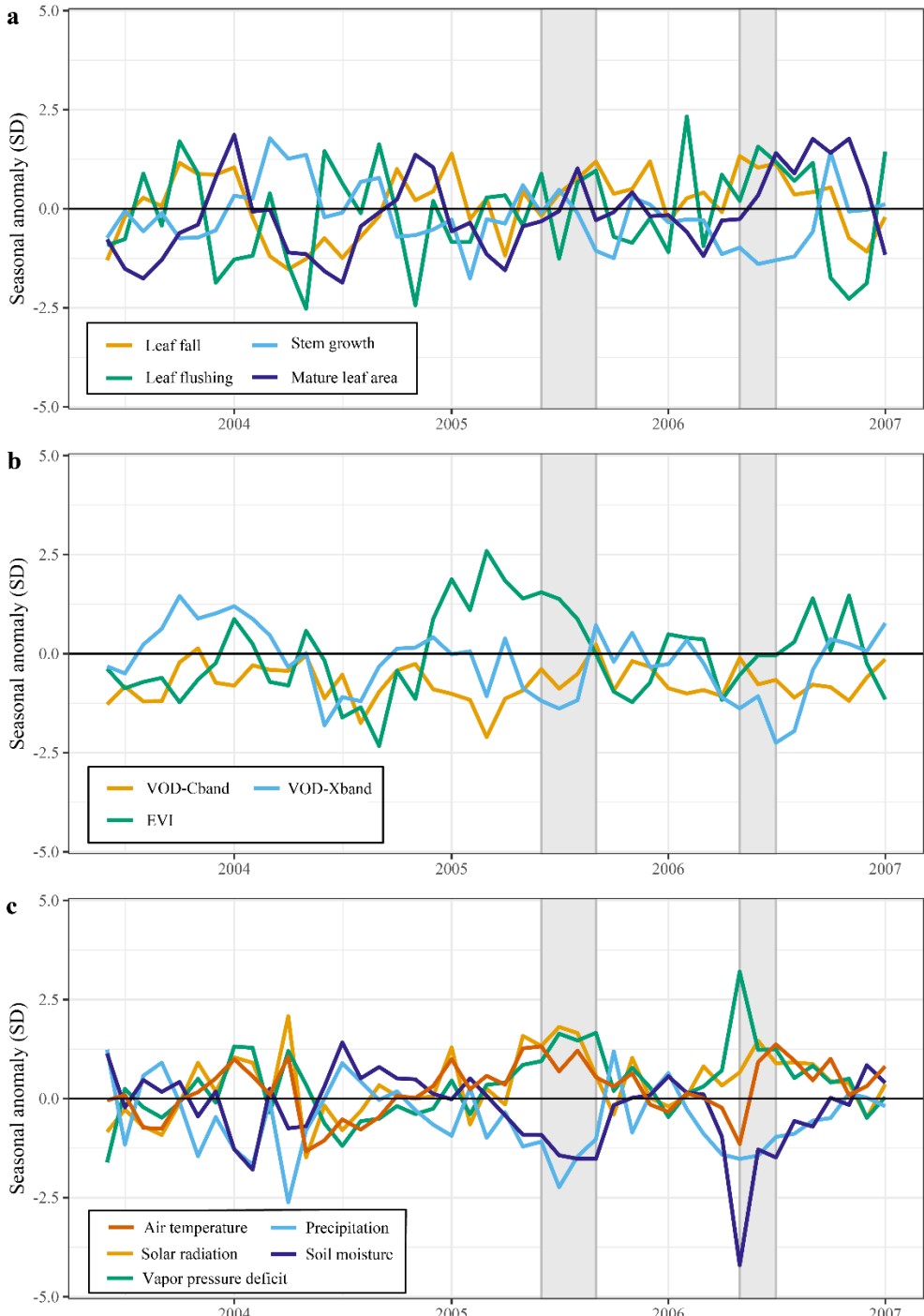

**Figure 8 responses of aboveground growth and remotely sensed vegetation properties to the 2005 drought (June – September 2005) and key climatic variables. All graphs show the trend in the standardized seasonal anomaly, the deviation from the monthly mean divided by the standard deviation of that month. Leaf fall and stem growth (a) are derived from the two separate XGBoost models providing monthly values from January 1982 until September 2019. Mature leaf area (a) is the sum of flushed leaves from 2 to 5 months in the past.**

### 3.5 Long-term trends and ENSO effects on aboveground growth

The long-term monthly estimates of stem growth and leaf litterfall were seasonally detrended (i.e. subtracting the monthly average to omit seasonality) to identify long-term trends and multi-year fluctuations in aboveground biomass production (Figure 9). The following statistics are derived from the timeseries that have been seasonally detrended and which have been smoothened using a moving average (Figure 9, black line). The seasonally detrended data suggests a significant increase of leaf production ($r = 0.61$, $p < 0.001$, $5.96 \cdot 10^{-3} \pm 0.37 \cdot 10^{-3}$ Mg C ha$^{-1}$ yr$^{-2}$) in the Amazon basin between 1982 and 2019 (Figure 9a). However, this increase in leaf litterfall is partly offset by a decline in stem growth in the same period ($r = -0.52$, $p < 0.001$, $-1.96 \cdot 10^{-3} \pm 0.15 \cdot 10^{-3}$ Mg C ha$^{-1}$ yr$^{-2}$, Figure 9b).

To more appropriately compare the empirically modelled trends in stem growth change to the trends in stem growth found in a network of forest plots across the Amazon basin (e.g. Brienen et al., 2015; Hubau et al., 2020), the modelled monthly stem growth values were also extracted for the locations of these inventory plots (Figure S6). Also at the locations of the inventory plots, the long-term model estimates show a significant decline in stem growth, very similar to the trend for the entire Amazon basin ($r = -0.58$, $p < 0.001$, $-2.26 \cdot 10^{-3} \pm 0.15 \cdot 10^{-3}$ Mg C ha$^{-1}$ yr$^{-2}$).

The significant decline of stem growth and increase of leaf litterfall over time in the Amazon basin is possibly driven by the warming and drying of the Amazonian climate. While surface air temperature was found to have increased between 1982 and 2019 ($r = 0.58$, $p < 0.001$, $1.97 \cdot 10^{-2} \pm 0.13 \cdot 10^{-2}$ °C yr$^{-1}$) top-soil volumetric moisture content declined ($r = -0.52$, $p < 0.001$, $-3.35 \cdot 10^{-4} \pm 0.26 \cdot 10^{-4}$ m$^3$ m$^{-3}$ yr$^{-1}$, Figure 9d). However, distinct regional differences are visible in the trends of leaf litterfall, stem growth, top-soil volumetric moisture content and vapour pressure deficit (Figure S7). While the central Brazilian Amazon shows a significant ($p < 0.05$) drying trend coinciding with a clear positive trend in leaf litterfall and a negative trend in stem growth, large areas within Suriname, Guyana and eastern Venezuela show regional wetting and also a significant increasing trend of stem growth.

Superimposed on the long-term trends is the short-term variability in leaf litterfall, stem growth and aboveground biomass production that seem strongly related to ENSO (Figure 9d). Here, the multi-variate ENSO index is used as a measure of ENSO phase and intensity (Wolter and Timlin, 2011). The strong coupling between ENSO and the empirically modelled leaf litterfall and stem growth rates is to be expected as the climate variables used to estimate leaf litterfall and stem growth are also strongly impacted by ENSO. Nonetheless, it is noteworthy that three major El Niño related droughts in 1997, 2010 and 2015 can be identified as periods with high temperatures, relatively low soil moisture, high leaf litterfall and low stem growth (Figure 9).

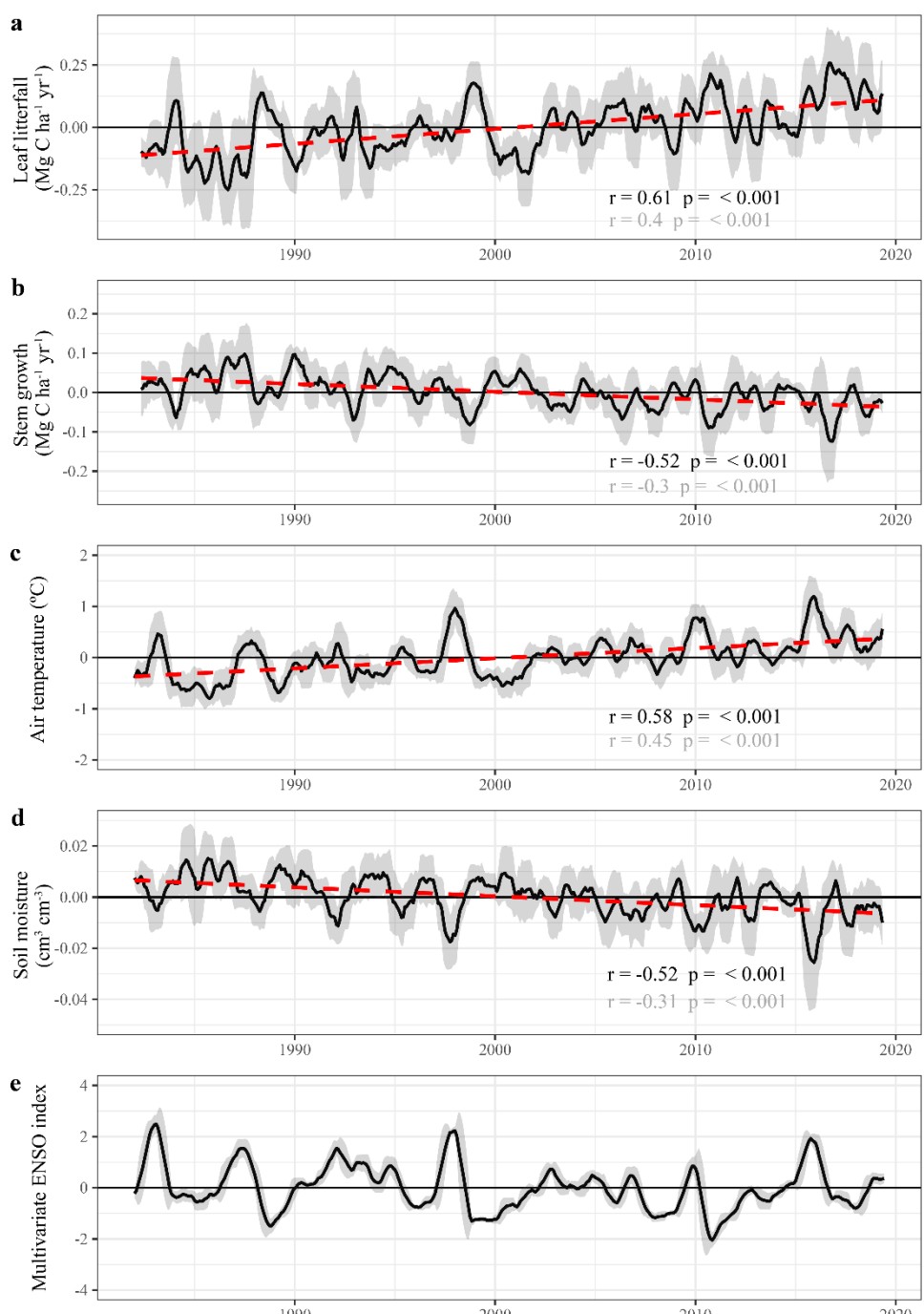

**Figure 9 Long-term predictions of seasonally detrended anomalies in aboveground growth, air temperature and soil moisture across the Amazon basin and the relation with the multivariate ENSO index (Wolter and Timlin, 2011). Black lines are the 9 month moving average of the anomalies and the dark grey uncertainty bands show the moving standard deviation of the same data. Red dashed lines represent the least squares linear regression fit through the averaged time-series. Test statistics are provided for both the linear** 515 **regression of the moving average (black) and the original monthly data (grey).**

## 4. Discussion

### 4.1 Drought effects on leaf phenology and canopy productivity in Neotropical forests

This study aimed to investigate how leaf litterfall, leaf flushing and stem growth change in response to drought in Amazonian forests. The long-term empirically modelled estimates of leaf fall showed that during the peak of the 2005 and 2015 droughts in the Amazon basin, leaf fall was significantly higher compared to its monthly averages in these months. Furthermore, estimated leaf fall was also elevated during other historical droughts in 1987, 1997 and 2009-2010 across the Amazon basin (Figure 9). These results confirm earlier site specific studies that reported elevated leaf litterfall during drought (Bonal et al., 2008; Rice et al., 2008; Roberts et al., 1990; Wieder and Wright, 2001) and during periods of warm and dry conditions associated with a strong El Niño event (Detto et al., 2018; Thomas, 1999). A straightforward explanation of the observed increase in leaf litterfall during drought is that leaf shedding directly reduces tree water use. Next to a progressive closure of the leaf stomata to limit transpiration, many tropical tree species are found to shed their leaves and thereby reduce the demand of water during drought (Wolfe et al., 2016). Therefore, leaf shedding in trees helps to limit transpiration during drought and maintain the hydraulic integrity of the water transporting tissue (Janssen et al., 2020; Wolfe et al., 2016). Although there is a large variability of drought avoidance and drought tolerance strategies within Amazonian tree species, with some trees maintaining transpiration and leaf area during drought (Bonal et al., 2000a; Brum et al., 2018, 2019; Janssen et al., 2020, 2019; Maréchaux et al., 2018; Oliveira et al., 2019), if a proportion of trees shed their leaves to avoid dehydration, all other things being equal, this will show up as increased leaf litterfall on the stand scale.

In contrast to leaf litterfall, the estimated seasonally detrended timeseries of leaf flushing showed positive anomalies in the early and final months of the 2005 and 2015 droughts (Figure 5a, 8a). Especially the pulse of newly flushed leaves in the early months of the 2015 drought resulted in above-average mature leaf area (i.e. the sum of leaf area flushed in the past 2-5 months) during the peak of the drought. A similar response of leaf flushing was observed during the short drought of 2006 (Figure 8). Leaf flushing at the onset of a drought is in apparent contradiction with the observed enhanced leaf shedding, which is presumably drought-induced to limit transpiration. However, these results can be reconciled as 1) the timing of enhanced leaf flushing at the onset and end of the drought was different from the peak in enhanced leaf litterfall during the height of the drought, and 2) leaf litterfall and leaf flushing often simultaneously occur in wet tropical forests, sometimes on the same tree or even the same branch (Borchert, 1994). Enhanced leaf flushing at the onset of a drought in wet forests can be explained by maintained water uptake through deep soil water access in the early months of a drought (Bonal et al., 2000c; Brum et al., 2019; Meinzer et al., 1999; Nepstad et al., 1994). Furthermore, leaf photosynthetic capacity declines with leaf age (Albert et al., 2018; Kitajima et al., 2002; Menezes et al., 2021) and the capacity of stomates to close under dry conditions also declines with leaf age (Reich and Borchert, 1988). Therefore, the shedding of old leaves and flushing of new leaves with high photosynthetic capacity and highly responsive stomates might be a suitable strategy for tropical trees to adopt during drought.

Those areas in the Amazon basin that experienced increased leaf flushing and showed a higher mature leaf area in 2005 and 2015 also showed higher values of the MODIS enhanced vegetation index (EVI). These results therefore corroborate the finding that during the 2005 and 2015 droughts, large areas within the Amazon basin showed a green-up, visible as positive anomalies in the MODIS EVI (Saleska et al., 2007; Yang et al., 2018). Furthermore, these findings support *in situ* observations that showed that leaf flushing was significantly enhanced at the end of the 2015 drought in the central Amazon, resulting in higher mature leaf area, associated with positive EVI anomalies in the year following the drought (Gonçalves et al., 2020).

However, enhanced leaf flushing, higher mature leaf area and positive anomalies in the MODIS EVI during the 2015 drought occurred mainly in ever wet forest of the central Amazon experiencing a short dry season (< 3 months). The eastern part of the basin that experiences a moderate to long dry season (> 3 months) (Sombroek, 2001), actually showed negative anomalies or no change in leaf flushing, mature leaf area and EVI (Figure 3a, 3d & 4c). These results suggest that leaf flushing and canopy green-up in response to drought only occurs in ever wet forests which do not experience a regular dry season.

Vegetation indices, such as the EVI and the normalized difference vegetation index (NDVI) are sensitive to vegetation chlorophyll content or "greenness" and have often been used to assess the effect of drought on the Amazon forest canopy. The earliest effects of droughts observed with satellites occurred during the 1983 and 1987 El Niño events, which caused negative anomalies in the NDVI from the NOAA Advanced Very High Resolution Radiometer (AVHRR) (Asner et al., 2000; Batista et al., 1997). However, a later El Niño related drought in 1997 resulted in positive AVHRR NDVI anomalies across the Amazon basin (Dessay et al., 2004). Furthermore, during the 2005 drought, positive anomalies in MODIS EVI were visible across the south-western Amazon, suggesting that the forest canopy greens-up in response to drought (Liu et al., 2018b; Saleska et al., 2007). This finding has been disputed and was attributed to insufficient atmospheric correction (Asner and Alencar, 2010; Samanta et al., 2010), sun-sensor geometry (Morton et al., 2014) and structural changes in the forest canopy (Anderson et al., 2010). However, our results suggest that the observed green-up during 2005 and especially the later 2015 drought might not be an artefact in the remote sensing data but an actual result of increased leaf flushing at the onset of drought.

It is noteworthy that droughts in which green-up has been observed (1997, 2005, 2006, and 2015) occurred during the second half of the year (June – December), which encompasses the dry season and early wet season in the eastern Amazon (Sombroek, 2001). Contrastingly, droughts in which no green-up was observed (1983, 1987, 2010) occurred predominantly in the first half of the year and therefore in the wet season. As leaf exchange in the Amazon basin occurs in the dry season, drought conditions might accelerate leaf flushing synchronous to the general phenology in the dry season but not in the wet season. That green-up during drought occurs despite the observed positive anomalies in leaf litterfall suggests that during drought, older leaves with lower photosynthetic capacity and higher near-infrared absorptance (Doughty and Goulden, 2009; Kitajima et al., 2002; Roberts et al., 1998) are shed, while newly flushed leaves are maintained. When taking into account the time that newly flushed leaves need to fully expand and attain their highest photosynthetic capacity, which is 2-5 months (Albert et al., 2018; Gonçalves

et al., 2020; Restrepo-Coupe et al., 2013), it can be argued that the observed green-up is not a direct effect of drought but rather a consequence of the environmental conditions at the onset of the drought.

Earlier studies hypothesized that increased incoming solar radiation during drought, as a result of a decline in cloud cover, might be driving the observed green-up (Saleska et al., 2007). Indeed, both spatial as well as temporal correlations between photosynthetic active radiation (PAR) and EVI were found in response to the 2015 drought (Yang et al., 2018). Lengthening of the photoperiod has been recognized as a key environmental cue for leaf abscission and flushing across evergreen tropical forests (Borchert et al., 2002, 2015; Elliott et al., 2006). Reduced cloud cover and increased direct solar radiation reaching the

forest canopy at the onset of an atmospheric drought, when soil water is still readily available, might therefore present an environmental cue for leaf flushing. This mechanism might explain the positive anomalies in leaf flushing observed at the onset of the 2005 and 2015 droughts (Figure 5a, 8a). Next to insolation, trees need to be well hydrated to enable cell expansion, bud break and consequently leaf flushing (Borchert et al., 2002). Also the presence of older leaves in the canopy can inhibit leaf flushing (Borchert et al., 2002). Therefore, the excessive shedding of older leaves during the height of the 2005 and 2015

droughts and tree rehydration following the first rain events (Figure S5) could have acted as a strong environmental cue for the second leaf flushing event that was observed at the end of the 2015 drought (Figure 5a & 8a) (Gonçalves et al., 2020).

### 4.2 Drought effects on stem growth in Neotropical forests

In contrast to the observed leaf flushing and leaf fall responses to drought, stem growth is significantly reduced in the drought areas of the Amazon basin during the 2015 drought and to a lesser extent in 2005 and 2006 (Figure 3c & 6c). Other historical

droughts in the Amazon basin in 1987, 1997 and 2009-2010 are clearly visible in the long-term estimates as periods of reduced soil moisture and reduced stem growth across the Amazon basin (Figure 9). These results generally confirm site specific studies that found significant stem growth reductions in Neotropical forests in response to drought. These include the 1997 and 2010 droughts in Costa Rica (Clark et al., 2003; Hofhansl et al., 2014), the 2008 drought in French Guiana (Stahl et al., 2010; Wagner et al., 2013), and the 2010 and 2015 droughts across the Amazon basin (van Emmerik et al., 2017; Feldpausch et al.,

2016; Rifai et al., 2018). The lack of a clear negative and long-term impact of the relatively short 2005 drought on estimated stem growth in the Amazon basin might explain why field observations failed to observed significant declines in stem growth during the 2005 drought (Phillips et al., 2009). The relative importance of drought duration, intensity and timing (wet season or dry season) in limiting stem growth in tropical forests remains unclear and the interactions between drought and local conditions (e.g. topography, water table depth, soil water holding capacity) still need to be disentangled.

Stem growth reductions in response to drought can be expected as tree water status and stem growth are tightly coupled. Firstly, stem wood and bark can store substantial amounts of water, which contribute 5-30% to daily water use in Neotropical tree species (Meinzer et al., 2003; Oliva Carrasco et al., 2015). About 50% of stem wood and bark volume consists of water which can in part be withdrawn during drought (Dias and Marenco, 2016; Poorter, 2008). The loss of water from elastic tissue can

result in a decline of stem growth or even a decline of stem girth (Baker et al., 2002; van Emmerik et al., 2017; Reich and Borchert, 1982; Stahl et al., 2010). These elastic changes in stem volume arising from changes in stem wood and bark water content do not represent actual changes in secondary growth. However, these elastic changes are often unintentionally present in dendrometer measurements and therefore also in our dataset. Secondly, tissue dehydration during drought can cause cell turgor loss in the vascular cambium, limiting cell division and therefore actual secondary growth (Borchert, 1994; Körner and Basel, 2013; Muller et al., 2011; Worbes, 1999). Therefore, it is reasonable to assume that water availability directly reduced stem growth during drought.

The long-term estimates of stem growth in this study point to a significant negative trend in stem growth in the Amazon basin between 1982 and 2019 (Figure 9b), which was not found in a basin-wide network of inventory plots for a similar timespan (1983-2011) (Brienen et al., 2015; Hubau et al., 2020). This is surprising as 60% of the data from these same inventory plots are used to train the stem growth model and were therefore expected to show similar long-term trends. As the plot scale data is very similar, this discrepancy has to be explained by the method of upscaling these plot scale observations. Firstly, the model provides stem growth estimates for more than 54 thousand grid cells covering the entire Amazon basin whereas the measured stem growth rates are measured at around 320 inventory plots scattered across the basin (Brienen et al., 2015). However, a spatial bias alone does not seem to be causing the discrepancy as a similar negative trend in estimated stem growth was visible at the locations of the inventory plots (Figure S6). Secondly, the majority of stem growth observations from inventory plots are from the 2000's and 2010's (Figure S3) while the model estimates go back to 1982. If this temporal bias is a factor causing the discrepancy, it suggests that stem growth in the inventory plots was underestimated in the 1980's and 1990's or that more productive plots were included in recent years. However, this temporal bias should have been corrected for in the trend analyses of Brienen et al. (2015) (see also Brienen et al,. 2015 Extended data Figure 3). Finally, as the model uses the ERA5 long-term reanalysis data of surface air temperature, precipitation and soil moisture to estimate stem growth, trends in the stem growth estimates are therefore reflecting the trends in the climate data (Figure 9). As stem growth in the Amazon basin generally declines in the dry season when soil moisture is low and air temperatures are high (e.g. Doughty et al., 2014; Girardin et al., 2016; Janssen et al., 2020a) a trend in soil moisture and temperature might therefore result in a predicted trend in stem growth which might not necessarily be reflecting the actual trend in stem growth. This would mean that the XGBoost models exaggerate the contribution of the changing climate variables on the long-term trends in stem growth. Therefore, data from tree census data from permanent inventory plots (Brienen et al., 2015; Hubau et al., 2020) is essential to be able to accurately model and upscale stem growth at multi-decal timescales

### 4.3 What are satellite sensors actually sensing?

The controversy surrounding the observation of Amazon canopy green-up during drought is mainly caused by differences in sensor sensitivity and the interpretation of the retrieved signals. Generally, canopy green-up is observed in multi-spectral remote sensing data during or following major droughts in the Amazon forest that are timed in or at the end of the regular dry

season (Gonçalves et al., 2020; Lee et al., 2013; Liu et al., 2018b; Saleska et al., 2007; Yang et al., 2018). Our results support this canopy green-up and attribute it to enhanced leaf flushing at the onset of a drought and subsequent leaf maturation in the following months (Figure 5a & 8a). However, canopy green-up does not necessarily have to result in, or be a consequence of, an increase in canopy photosynthesis or gross primary productivity. Indeed, *in situ* leaf scale photosynthesis is generally observed to decline during drought in Neotropical forests (Bonal et al., 2000b; Doughty et al., 2014; Janssen et al., 2020; Stahl et al., 2013). This is confirmed by satellite observations of negative anomalies in sun-induced fluorescence during drought (see also Figure 5b), which is considered a proxy of canopy photosynthesis (Koren et al., 2018; Lee et al., 2013; Yang et al., 2018). The observed decline in leaf-level photosynthesis during the 2015 drought in the central Amazon has been attributed to progressive stomatal closure and not to changes in leaf chemistry (Santos et al., 2018). These results suggest that despite canopy green-up, photosynthesis might well be downregulated during drought because of stomatal limitations (Janssen et al., 2020; Santos et al., 2018).

The analysis of changes in X band vegetation optical depth (VOD) in the area affected by drought in 2005, 2006 and 2015 (Figures 4d & 7d) confirms earlier results from passive and active microwave remote sensing studies that showed negative anomalies of VOD and radar backscatter in response to historical droughts in the Amazon basin (van Emmerik et al., 2017; Frolking et al., 2011, 2017; Lee et al., 2013; Liu et al., 2013, 2018b; Saatchi et al., 2013). During the 2015 drought in the central Amazon, van Emmerik *et al.* (2017) found that remotely sensed $K_u$ band radar backscatter declined during the drought which was strongly correlated to *in situ* observed declines in stem girth. In contrast to vegetation indices from multi-spectral remote sensing, passive and active microwave remote sensing is generally sensitive to vegetation biomass and water content and not vegetation greenness (Frappart et al., 2020; Liu et al., 2013; Meesters et al., 2005). Furthermore, X band VOD has been found to be strongly dependent on leaf water potential in temperate forests in North America (Momen et al., 2017). Therefore, the negative anomalies in VOD and radar backscatter in response to drought are likely signalling a decline in vegetation water content during drought (Momen et al., 2017) and can therefore be used as a rough proxy of tree water status and stem growth. Remotely sensed data can be extremely useful in identifying vegetation responses to extreme events like droughts on large spatial scales. However, as sensors are sensitive to different vegetation properties, the interpretation of observed responses should always be done with utmost care and preferably in a multi-sensor comparison.

## 5. Conclusions

Long-term monthly estimates of stem growth, leaf fall and flushing indicate that Amazon green-up during drought is a legacy effect of enhanced leaf flushing at the onset of a drought and cannot be considered a proxy of canopy photosynthesis, aboveground biomass production or forest health in evergreen Neotropical forest. Separating photosynthesis, vegetation water status and canopy greenness as three sometimes independent properties of the vegetation allows for explaining apparent discrepancies in drought responses visible in remote sensing data (e.g. Lee et al., 2013; Liu et al., 2018b). To exemplify,

Anderson *et al.* (2010) found that areas that showed the highest EVI green-up during the 2005 drought also experienced the highest rates of drought-induced tree mortality. Our results confirm that drought stress induced reductions in stem growth often coincide with enhanced leaf flushing and canopy greening during drought, which are not necessarily physiologically contradicting.

Our results also point to a long-term (1982-2019) decline in stem growth rates across the Amazon basin, which appears to be driven by increased warming and drying of the Amazonian climate. While still uncertain, this decline of carbon sequestration in woody stem growth over time ($-1.96 \cdot 10^{-3} \pm 0.15 \cdot 10^{-3}$ Mg C ha$^{-1}$ yr$^{-2}$) is significantly less compared to the trend of increasing carbon release through tree mortality ($25.5 \cdot 10^{-3}$ Mg C ha$^{-1}$ yr$^{-2}$) found in a network of forest inventory plots (Brienen et al., 2015). As tree mortality is elevated during and following drought (Feldpausch et al., 2016; Phillips et al., 2009) it is of critical
importance to study the drivers of drought sensitivity and drought-induced tree mortality in tropical forests to be able to project future changes in the carbon sink strength of the Amazon basin.

**Code availability**

The code used in this study is available upon reasonable request.

**Data availability**

The dataset compiled and analysed in this study will become publicly available on DataVerse NL (https://dataverse.nl/dataset.xhtml?persistentId=doi:10.34894/LY77IN) after the final revision and publication of the manuscript.

**Author contribution**

TJ conceived and designed the analysis, collated the dataset, carried out the analysis and wrote the article. YV and BD co-
designed the analysis and have critically reviewed and commented on the article. FH, SL, KN, KF and HD helped with the writing of the article and have critically reviewed and commented on the article.

**Competing interests**

The authors declare that they have no conflict of interest.

## Acknowledgements

We would like to thank all the individual researchers that enabled this study by providing freely available datasets and accessible figures in addition to their published work. Furthermore we would like to thank Chi Chen for providing us with the most recent version of the Global Data Set of Vegetation Leaf Area Index (LAI3g). We would also like to thank Maurits Kooreman for providing us with the Sun-Induced Fluorescence of Terrestrial Ecosystems Retrieval V2 dataset.

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

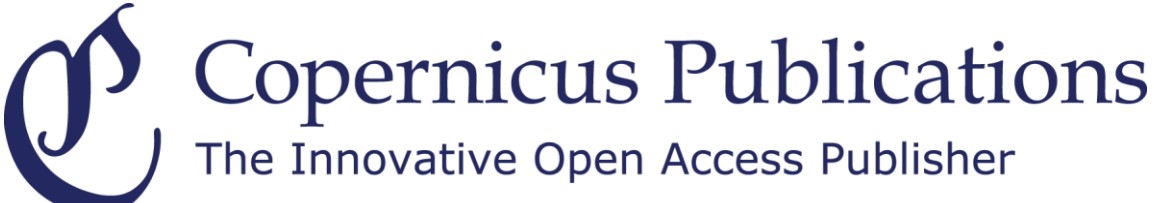

**Figure 1: The logo of Copernicus Publications.**