# Peer review of "Drought effects on leaf fall, leaf flushing and stem growth in the Amazon forest; reconciling remote sensing data and field observations"

_Biogeosciences, 2021_

## Author Comment (AC1)

**Response to referee #1**

Thank you very much for your elaborate review of our manuscript. We appreciate the time and effort put into this extensive review and your in-depth comments. We also very much appreciate the recognition of our work as a potential important contribution to the literature on forest growth responses to drought. Below, we will address your questions and comments individually in four sections.

The first question is about the time span of the ERA5 climate data and the remote sensing data (e.g. MODIS EVI) being different and how we dealt with this difference in the models. The answer is that we did not use the remote sensing data in the models as only the ERA5 climate data was used as a product with a temporal dimension (see also next section in this response). Besides the different time spans being an obstacle for using the remote sensing data in the model, keeping the remote sensing data out of the models also ensured that we could more confidently compare the remote sensing data and model predictions without lapsing into circular reasoning. In the revised version of the manuscript we will make sure that this will be explained more explicitly in the methods section.

Concerning the second issue raised, about using variables without a temporal dimension, such as soil properties, plant traits (e.g. SLA) and biomass, we acknowledge that these properties are not constant. We also agree that this might become a problem when these properties have dramatically changed in the past decades such as due to deforestation. This is a limitation of our analysis that deserves to be discussed more elaborately in the discussion section. However, as the referee acknowledges, the products used often provide only one value (e.g. SLA) without a time dimension and others are indeed not available when going back to the 1980's and 1990's, such as the high resolution biomass maps. Furthermore, when checking the site locations in our database prior to the analysis, we found that most if not all of the sites are (still as of 2020) located in protected areas and nature reserves. Visual inspection of these sites using recent aerial images (e.g. Google Earth) also shows that these inventory sites, while sometimes located in a very small forest fragment, seem to be relatively undisturbed. Nonetheless, for the revised version of the manuscript we have looked into the data sets recommended by the referee (Hansen et al. 2013 or Song et al. 2018) and included the long-term vegetation continuous field (VCF) dataset (1982-2019) from Song et al. 2018 into our analysis. In both the new leaf litterfall as well as the new stem growth models, the VCF was chosen as one of the 20 variables used in the final models. Therefore, it is likely explaining some of the temporal and spatial variability in stem growth and leaf litterfall data that was not explained by the ERA5 climate data and other geospatial datasets. We appreciate your suggestion of including the VCF dataset in our analysis.

We agree that our current validation approach does not validate model performance across the sites and that we do not currently know how well the models perform over space alone, only over space and time combined. Therefore, we welcome the idea from the referee to include a second validation of model performance by splitting the test and training data based on the sites and see how well the model performs at the sites not included in the training data. To test how well the model performs across the sites, we have trained the model using the predefined selected features and the hyperparameters (from the existing model) but now only using training data consisting of measurements from 60% of the sites in the dataset. The model evaluation on the data from the remaining 40% of the sites is presented (Figure 1) and will be included in the revised

supplement. As expected, the model performance declined in both the stem growth as well as the leaf litterfall models when using only 60% of the sites, more so in the leaf litterfall model than the stem growth model. The absolute (RMSE) and relative error (NRMSE) increased only by about 0.01 to 0.04 Mg C ha-1 month-1 and 1 to 4.4 percentage points, respectively. However, the explained variability (R2) did decline significantly from 0.5 to 0.4 in the stem growth model and from 0.67 to 0.38 in the leaf litterfall model. These results suggest that the model performs better when explaining mainly the temporal variation in leaf litterfall and stem growth compared to mainly the spatial variation. The results also suggest that the model performs better in predicting the spatial variation in stem growth compared to leaf litterfall, which might indicate that some of the drivers of spatial variability in leaf litterfall are not well represented in our current model.

Figure 1 Model evaluation of the new site based validation. The scatterplots show the predicted biomass production versus the measured biomass production of the test data (40% of the field sites) that was used to validate the stem growth (a) and leaf litterfall (b) models. The dashed black line is the 1:1 line and the dashed red line the least squares linear regression fit.

The XGBoost model unfortunately does not offer a way to easily obtain uncertainties from the model output. The structure of the model, which is a sequence of regression trees and not a group of separate regressions trees that are averaged like in a random forest model, does not allow for a standard uncertainty estimate. There are to our knowledge a few workaround techniques to still obtain uncertainty estimates from XGBoost models. One of these workarounds is to train a separate XGBoost model to predict the error in the original test dataset (see https://medium.com/@gucit/a-simpletechnique-to-estimate-prediction-intervals-for-any-regression-model-2dd73f630bcb). We employ this method in the revised version of our manuscript to derive a measure of uncertainty for the stem growth and leaf litterfall estimates (Figure 2). The results of this new uncertainty analysis are depicted in Figure 2 and will be included in the new supplement accompanying our revised manuscript. The figure shows that the absolute error (RMSE) of the stem growth and leaf litterfall models is low in the high elevation ecosystems of the Andes compared to the lowlands (Figure 2a, 2b). However, when the RMSE is normalized using the average seasonal range in stem growth and leaf litterfall values (annual amplitude) the opposite pattern is observed (Figure 2c, 2d). In this case, the relative error for both the stem growth and leaf litterfall models is very high in the

Andes region (> 40%) compared to the lowland forest of the Amazon basin, the Cerrado, Caatinga, and Atlantic forest regions and central America (

---

## Author Comment (AC2)

**Response to referee #2**

Dear referee,

Thank you very much for your kind review. The time and effort put into the reading of our manuscript and writing this review is greatly appreciated. Furthermore, thank you for recognizing our work as a potential useful contribution to the existing literature. Below, we will address your major concerns and questions.

First, we agree that while our dataset covers the entire neo-tropics, the analysis is focused specifically on the Amazon basin. Therefore, it would indeed be appropriate to change the title into: "Drought effects on leaf fall, leaf flushing and stem growth in the Amazon basin; reconciling remote sensing data and field observations".

Considering the imbalance of leaf litterfall and stem growth observations over time in the dataset, it is true that the density of observations changes over time and that this might impact the model uncertainties as well. We now show these trends in a new supplementary figure (Figure 1). In general, there was an increase in observations of both leaf litterfall and stem growth in the 2000's compared to the 1980's and 1990's (Figure 1a). However, since 2010 there has been a steady decline in the number of observations per year which can at least partly be explained by a larger contribution of data that is presently not published or under embargo. There is also a positive trend in the normalized root mean squared error (RMSE) of the leaf litterfall and stem growth model estimates (Figure 1b) suggesting that the relative model error increases over time in both models. For stem growth, the increasing trend in NRMSE is mainly driven by the decline of the averaged predicted stem growth in the dataset while the absolute error did not change over time (Figure 1c). In the case of leaf litterfall, the increase in NRMSE over time is driven by the increase in absolute error (Figure 1c) despite the increase of average leaf litterfall over time. This suggests that despite more data being available in recent year, the model error is actually higher compared to the 1980's and 1990's. We acknowledge that these issues were not specifically discussed in our manuscript. In the revised manuscript we will examine these uncertainties in more detail. Furthermore, the dataset compiled for this study will become public in the final stages of the publication process.

[Figure]

*Figure 1 New supplementary figure showing the change in leaf litterfall and stem growth observations in the dataset and the error of the leaf litterfall and stem growth estimates over time. The normalized root mean squared error (RMSE) is calculated as the RMSE divided by the predicted average stem growth and leaf litterfall.*

We have "cropped" the model output for the analyses to cover the extent of the Amazon basin. This was done because the focus of the study was on the drought related changes in forest growth that occurred in the Amazon basin. Furthermore, the figures were cropped to cover the area of interest as local differences cannot be identified if the entire study area was shown (details are lost when "zooming out"). Furthermore, as can be observed in for example Figure 3 and briefly discussed in the results (L345-348), different responses in the modelled stem growth and leaf litterfall and the remote sensing data are shown for the humid forests of the Amazon basin and the more dry Cerrado and caatinga regions. It would have made the paper considerably lengthier if we had to

examine the drought-induced changes that occurred in every region or ecosystem covered by the model output. However, we agree that these choices have not been explicitly mentioned in the introduction or methods sections of the manuscript. We will highlight our choices considering the region of interest in the revised version of the manuscript.

Thank you for  noting the significant negative anomaly in soil moisture and positive anomaly in VPD in April 2006 that can be observed in Figure 8 of the manuscript and the associated negative anomalies in stem growth in the same period. We believe that this can be considered a short drought period, which indeed seems more anomalously dry compared to the 2005 drought. This period was not mentioned in the results of the current version of the manuscript as there is to our knowledge no mention of this 2006 drought event in the literature. One explanation for this is that this drought occurred in the middle of the wet season in the western Amazon so that it was probably not recognized as a drought and did not result in a significant physiological impact on the vegetation. However, we agree that this period should at least be mentioned in the results and we will discuss this 2006 event in the revised version of the manuscript.

We agree that in the conclusion we do not reveal our opinions about whether remote sensing and ground-based trends can be reconciled and whether ground based carbon sequestration is overestimated. What we aimed to communicate in the discussion is that remote sensing techniques differ considerably in what they measure and how these measurements are related to forest growth and "health". For example, we show that canopy greening is not a good proxy of above-ground growth or drought stress in evergreen tropical forests. Furthermore, we show that ground-based estimates can be upscaled using climate data and other geospatial datasets to obtain a different temporal trend in stem growth compared to when the measurements of largely the same inventory plots are averaged. It would be interesting for future research to examine which upscaling technique provides the most accurate trend estimation, but for now we cannot confidently conclude which technique is more appropriate. In the revised version of the manuscript, we will more elaborately mention these issues and considerations in the conclusion.

Thank you again for your review and we hope that in our response we have addressed all your major concerns and questions regarding our manuscript. We will carefully address all your minor and major comments when revising our manuscript, which will definitely enhance the quality of the revised manuscript.

On behalf of my co-authors,

Thomas Janssen

---

## Author Response (AR1)

**Reply to referee #1**

**General comments**

Understanding how tropical forest trees respond to extreme events is critical if we want to quantify the carbon cycle of the terrestrial biosphere to ongoing climate change. In this study, Janssen and co-authors use a collection of remote sensing and gridded environmental variables to develop machine learning models to predict leaf liter fall and stem growth across the tropical and sub-tropical areas of the Americas. The authors used data retrieved from the literature to train and test the models, and applied the model over long time series to identify how leaf production and stem growth during Amazonian droughts and to explore the long-term trends, and found increased leaf production but reduced stem growth during droughts, and a small but significant long-term decline in stem growth.

Overall, this is a very well-written paper with a mostly clear and interesting analysis that integrates historic observations with multiple remote sensing products. It has a potential to become an important contribution to advance our understanding of two critical processes in forest dynamics (leaf litter fall and stem growth) with limited information across the tropics beyond intensively studied sites. However, I have some questions and points, mostly regarding the machine learning methods, that may deserve additional clarification and potential development to assess and improve the model robustness (see below).

*Thank you very much for your elaborate review of our manuscript. We appreciate the time and effort put into this extensive review and your in-depth comments. We also very much appreciate the recognition of our work as a potential important contribution to the literature on forest growth responses to drought. Below, we will address your questions and comments individually point by point.*

First, unless I missed it, the authors did not explain how they addressed the issue of some time series being shorter than others. For example, most meteorological drivers extend from 1981 to 2019, whilst MODIS EVI is not available before 2000. What was exactly done in the model for the periods in which less data were available?

*We did not use the remote sensing data in the models as only the ERA5 climate data was used as a product with a temporal dimension (see also next section in this response). Besides the different time spans being an obstacle for using the remote sensing data in the model, keeping the remote sensing data out of the models also ensured that we could more confidently compare the remote sensing data and model predictions without lapsing into circular reasoning and interdependencies. In the revised version of the manuscript these considerations are now highlighted in the methods section (new L159-160).*

In addition, the authors used a single time for a few variables, and presumably assumed the values constant (if not, please clarify). I understand that some data sets may be simply not available for more than one time (SLA) or their time series may be uncertain (biomass), but assuming these quantities constant is a substantial simplification for the time span of this study. This is especially true for deforested areas and forest edges, which may rapidly change. There are a few data sets that could work as proxies for the forest dynamics (for example, Hansen et al. 2013 or Song et al. 2018 for tree cover).

*We acknowledge that these properties are not constant. We also agree that this might become a problem when these properties have dramatically changed in the past decades such as due to deforestation. However, as the referee acknowledges, the products used often provide only one value (e.g. SLA) without a time dimension and others are indeed not available when going back to the 1980's and 1990's, such as the high resolution biomass maps. Furthermore, when checking the site locations in our database prior to the analysis, we found that most if not all of the sites are (still as of 2020) located in protected areas and nature reserves. Visual inspection of these sites using recent aerial images (e.g. Google Earth) also shows that these inventory sites, while sometimes located in a very small forest fragment, seem to be relatively undisturbed. Nonetheless, for the revised version of the manuscript we have looked into the data sets recommended by the referee (Hansen et al. 2013 and the updated version of Song et al. 2018) and included the long-term vegetation continuous field (VCF) dataset (1982-2019) from Song et al. 2018 into our analysis. In both the new leaf litterfall as well as the new stem growth models, the VCF was chosen as one of the 20 variables used in the final models (see updated selected features in the results section 3.1, final paragraph). Therefore, it is likely explaining some of the temporal and spatial variability in stem growth and leaf litterfall data that was not explained by the ERA5 climate data and other geospatial datasets. We appreciate your suggestion of including the VCF dataset in our analysis.*

The authors used data digitised from published figures, which is truly a heroic effort, yet the sample is spatially very limited, and this makes me wonder how robust the model is for spatial extrapolation. When the authors tested the model performance, they retained 60% of data from each site for training, and used the remaining 40% of the data for cross-validation. This approach tests how well the model performs in each site, but it does not tell the accuracy of the model predictions in grid cells with no data. I suggest the authors to perform an additional test in which the training/testing data are split by sites (i.e., no data from the test sites are provided to the machine learning during training). This will be an imperfect assessment as some regions do not have any data (e.g., stem growth in the Caatinga region), but at least it may indicate some of the model limitations more clearly.

*We agree that our current validation approach does not validate model performance across the sites and that we did not show how well the models perform over space alone, only over space and time combined. Therefore, we welcome the idea from the referee to*

*include a second validation of model performance by splitting the test and training data based on the sites and see how well the model performs at the sites not included in the training data. To test how well the model performs across the sites, we have trained the model using the predefined selected features and the hyperparameters (from the existing model) but now only using training data consisting of measurements from 60% of the sites in the dataset.*

*The model evaluation on the data from the remaining 40% of the sites is presented in the results section 3.1, first paragraph, and is now included in the revised supplement (new Figure S2). As expected, the model performance declined in both the stem growth as well as the leaf litterfall models when using only 60% of the sites, more so in the leaf litterfall model than the stem growth model. The absolute (RMSE) and relative error (NRMSE) increased only by about 0.01 to 0.04 Mg C ha$^{-1}$ month$^{-1}$ and 1 to 4.4 percentage points, respectively. However, the explained variability (R$^2$) did decline significantly from 0.5 to 0.4 in the stem growth model and from 0.67 to 0.38 in the leaf litterfall model. These results suggest that the model performs better when explaining mainly the temporal variation in leaf litterfall and stem growth within sites using incomplete time series as training data compared to mainly the spatial variation between sites using complete time series as training data. The results also suggest that the model performs better in predicting the spatial variation in stem growth compared to leaf litterfall, which might indicate that some of the drivers of spatial variability in leaf litterfall are not well represented in our current model.*

Finally, I can see the maps generated in this study to be used by many other studies, and I think the authors could think of ways of spatially quantifying the uncertainties associated with the predicted quantities.  For example, I imagine that the XGBoost approach generates multiple predictions (from each regression tree) that could be used to estimate the uncertainties for each time and location.  There are also uncertainties in the training data sets that can be incorporated into the total uncertainty (somewhat similar to Chave et al. 2004), although this may not be feasible here because of the lack of uncertainty from the digitised observations.

*The XGBoost model unfortunately does not offer a way to easily obtain uncertainties from the model output. The structure of the model, which is a sequence of regression trees and not a group of separate regressions trees that are averaged like in a random forest model, does not allow for a standard uncertainty estimate. There are to our knowledge a few workaround techniques to still obtain uncertainty estimates from XGBoost models. One of these workarounds is to train a separate XGBoost model to predict the error in the original test dataset (see [https://medium.com/@qucit/a-simple-technique-to-estimate-prediction-intervals-for-any-regression-model-2dd73f630bcb](https://medium.com/@qucit/a-simple-technique-to-estimate-prediction-intervals-for-any-regression-model-2dd73f630bcb)). We employ this method in the revised version of our manuscript to derive a measure of uncertainty for the stem growth and leaf litterfall estimates (new Figures S3 and S4). The results of this new uncertainty analysis are depicted in the new supplementary Figure S4 accompanying our revised manuscript. The figure shows that the absolute error (RMSE)*

*of the stem growth and leaf litterfall models is low in the high elevation ecosystems of the Andes compared to the lowlands (Figure S4 a, b). However, when the RMSE is normalized using the average seasonal range in stem growth and leaf litterfall values (annual amplitude) the opposite pattern is observed (Figure S4 c, d). In this case, the relative error for both the stem growth and leaf litterfall models is very high in the Andes region (> 40%) compared to the lowland forest of the Amazon basin, the Cerrado, Caatinga, and Atlantic forest regions and central America (< 20%). These results suggest that the performance of both models is relatively weak in mountainous ecosystems, presumably due to large differences in climate and soils on relatively short spatial scales, compared to the other ecosystems included in the analysis. The results of the uncertainty analysis are now presented in the manuscript, in the results section 3.2, third paragraph. Thank you for your suggestion to perform an uncertainty analysis as it will increase the usability of our model output.*

*Thank you again for your elaborate review and we hope that with this response we have addressed your most pressing concerns and questions regarding our manuscript. We are confident that addressing your major and minor comments in the revised version of the manuscript, the quality of the manuscript will be greatly enhanced. Below we will reply to the specific comments.*

**Specific comments**

Domain: The authors used data from 30°S to 30°N (L. 115), but most figures show the results between 20°S and 20°N, sometimes including Eastern South America and sometimes excluding it.  Considering that most of the data are from tropical forests and the discussion is focused on the Amazon, I wonder if the domain should be consistently defined as the Amazon or moist tropical forests only.

*Yes, the paper focusses mainly on the changes in stem growth and leaf litterfall observed in the Amazon basin. After a similar comment from referee #2 we have changed the title of the manuscript to: "Drought effects on leaf fall, leaf flushing and stem growth in the Amazon forest; reconciling remote sensing data and field observations". Furthermore, changes have been made in the aims of the study (L109-L111) to highlight that the focus of the manuscript is on the Amazon region.*

Section 2.3.  I think this section needs more information about the processing of the remote sensing data. For example, MODIS data sets come with multiple quality flags, and the results can be highly influenced by the choices on how data were filtered and aggregated. I suggest explaining this processing either in the section or as a supporting information.

*Thank you for noticing this lack of clarification in the methods. We have now included more information about the pre-processing of the MODIS product. The only two pre-processing steps that were taken, were the masking of unreliable pixels using the pixel reliability layer (now included in L198-L199) and the monthly averaging (L199-L200). For*

*the other remote sensing products, no more pre-processing was done then already mentioned in the methods as most products are already extensively pre-processed and validated. We hope that this additional explanation of the MODIS product pre-processing and quality control is sufficient.*

Figure 1.  The model predictions show a pattern that is rather common in machine learning regressions (overestimation at the lower range, and underestimation at the upper range). In other words, the tails are biased.  This presumably affects the predictions at the extreme events (droughts), when one would expect litter fall and stem growth to be anomalous too.  I think this limitation must be highlighted when presenting and discussing the results.

*We agree that this is a limitation and discuss this problem more elaborately in the revised version of the manuscript in the results section (L270-L276).*

Section 4.1. I am somewhat confused with the mechanisms the authors are describing in this section, and they seem contradictory as presented now.  The authors mention in the first paragraph that plants may shed leaves to maintain the xylem integrity, but then in the second paragraph they show large positive anomalies in the early months of the drought.  Wouldn't this greening lead to higher transpiration rates and higher risk?

*Yes, this section was confusing and contradicting mechanisms were discussed without a clear mention of these contradictions. In the revised version of the manuscript we have included five sentences in the second paragraph of this section in which the contradiction of enhanced leaf flushing and leaf litterfall is discussed (L472-L482). Our explanation is as follows: Leaf flushing at the onset of both droughts is in apparent contradiction with the observed enhanced leaf shedding, which was presumably drought-induced to limit transpiration. However, these results can be reconciled as 1) the timing of enhanced leaf flushing at the onset and end of the drought was different from the peak in enhanced leaf litterfall during the height of the drought, and 2) leaf litterfall and leaf flushing often simultaneously occur in neotropical forests, even on the same tree (Borchert, 1994). Enhanced leaf flushing at the onset of the drought can be explained by maintained water uptake through deep soil water access in the early months of the drought, often observed in Neotropical forests (Bonal et al., 2000b; Brum et al., 2019; Meinzer et al., 1999; Nepstad et al., 1994). Furthermore, leaf photosynthetic capacity declines with leaf age (Albert et al., 2018; Kitajima et al., 2002; Menezes et al., 2021) and the capacity of stomates to close under dry conditions also declines with leaf age (Reich and Borchert, 1988). Therefore the shedding of old leaves and flushing of new leaves with high photosynthetic capacity and highly responsive stomates might be a suitable strategy for tropical trees to adopt during drought.*

L538.  Presumably, these trends were also present in the environment where measurements were carried out. In this case, either the ERA5 trends are stronger than observed, the XGBoost model exaggerated the contribution of climate to stem

growth and leaf litter fall, or the climate trends were stronger in areas where the authors did not have any training data. I think the interpretation of the discrepancy between the model and the reference data must be discussed more clearly.

*Thank you for your insights into this discrepancy, we have adapted the analysis to partly omit the spatial bias in the forest plots (Figure S6) and we have changed this paragraph to be more clear about the possible underlying causes of this discrepancy.*

**Minor comments**

L23. Briefly mention what are the other geospatial datasets.

*This has been changed to: "... and other geospatial datasets (various soil, terrain and vegetation properties) as explanatory variables."*

L47. Both wet and dry extremes are becoming more recurrent in the Amazon (Gloor et al. 2013).

*Thank you for the suggestion, this has been changed to: "Additionally, the Amazon region is experiencing an intensification of the hydrological cycle with increasing wet season precipitation, declining dry season precipitation, more frequent episodic droughts and increasing regional air temperatures (Cox et al., 2008; Fu et al., 2013; Gloor et al., 2013; Janssen et al., 2020a; Jiménez-Muñoz et al., 2016)."*

L69. There have been also studies suggesting that the apparent green-up could be an artifact caused by sun sensor geometry (Morton et al. 2014).

*This reference has now been included in a new sentence (L69-L71): "However, the apparent green-up during drought has been attributed to changes in atmospheric properties during drought (Asner and Alencar, 2010; Samanta et al., 2010), coined as an artefact of sun-sensor geometry (Morton et al., 2014) and resulting from structural changes in the forest canopy (Anderson et al., 2010)."*

L199. State somewhere in this paragraph the time span of the SIF data.

*Yes, this has now been included (L208): "Monthly images were available from January 2007 until December 2016."*

L205. Also indicate the time span of the VOD data.

*This has now been changed to: "We used C band (June 2002 – December 2018) and X band (December 1997 – December 2018) VOD data from the global long-term Vegetation Optical Depth Climate Archive (Moesinger et al., 2020)." (L218-L219).*

L232. Is there any justification for the 3-month window?

*The justification has now been added: "The 3 month window size, the lowest possible window size, was chosen to reduce the sometimes large month to month variation in leaf litterfall and stem growth while maintaining a large proportion of the within year variation to identify extremes." L240 – L242*

L323. At least from Fig. 3, I do not see consistent increase in leaf flushing across the boxed region, I see both positive and negative anomalies.

*We agree, there are both positive and negative anomalies in leaf flushing visible in Figure 3. Leaf flushing has been removed from this sentence.*

L328. Why not showing air temperature? This could appear in the Supporting Information.

*Air temperature has been added to the plots.*

L336. This could also appear in the Supporting Information.

*This has now been included in the Supplement (Figure S5).*

L347. Use "drier" instead of "dryer" (check for additional occurrences)

*Thank you, all mentions of dryer in the manuscript have been replaced by drier.*

L347.  The EVI response in central Amazon does not look like a consistent green-up. There is a strong negative anomaly band in the central part of the box.  This seems to coincide with an area thought to more susceptible to droughts (Hirota et al. 2011; Longo et al. 2018), and that was also affected by significant fires during the 2015-2016 drought (Aragão et al. 2018; Withey et al. 2018).

Thank you for noticing this interesting pattern, we have now highlighted this contrast between the wet west and dry east of the bounding box in the methods: *"However, also a belt shaped region of negative anomalies in EVI, leaf flushing and mature leaf area is visible in the drought area, going from the south-east to north-west across the bounding box (Figure 3b, 3d & 4c). This area experiences a relatively long dry season (≥ 4 months) compared to the forest in the west (< 3 months), suggesting that ever wet forests green-up during drought while seasonally dry forests do not (Sombroek, 2001)."*

*We also shortly discuss this contrast in the discussion: "However, enhanced leaf flushing, mature leaf area and EVI during the 2015-2016 drought occurred mainly in ever wet forest experiencing a short dry season (< 3 months) and high annual precipitation. The forest experiencing a moderate to long dry season (> 3 months) (Sombroek, 2001) actually showed negative anomalies or no change in leaf flushing, mature leaf area and EVI (Figure 3a, 3d & 4c). These results suggest that leaf flushing and canopy green-up in*

*response to drought only occurs in ever wet forests which do not experience a regular dry season."*

L385.  There are some significant positive anomalies of VOD along the rivers. It is hard to tell but it looks like the authors did mask inland (permanent) water bodies (if not, I suggest doing it). So it is a bit puzzling to me why the areas near the rivers would show such strong increase in 2005, when river levels were so low (Tomasella et al. 2011).

*We believe that this is an artefact of the VOD data caused by flooding. Our hypothesis is that when there is flooding near the tributary rivers of the Amazon, this will result in negative anomalies in VOD because of some issues in the algorithm retrieving VOD from the passive microwave signal. This will result in positive anomalies in VOD when there is no flooding and river levels are low as was the case in 2005 and 2015. The discussion of these issues with the VOD product extent beyond the scope of our study.*

L391. From Fig. 8c, it looks like 2006 was more extreme than 2005 itself. Also, from Fig. 8b, it looks like EVI was already very positively anomalous even before the drought.

*Yes, we have now included a mention of this short but intense 2006 drought in the results.*

L418. Did the authors consider dividing the values by the monthly standard deviation? It may help to remove the seasonal variation of the interannual variability still visible in Fig. 9.

*Yes, we also tried dividing the anomalies by the monthly standard deviation but this did not reduced the interannual variability. Furthermore, we believe that using the absolute anomalies (in Mg C ha$^{-1}$ year$^{-1}$) instead of the standard deviation helps with the interpretation of the trends in the figures.*

L436 (and in other places throughout the text).  El Niño is the warm phase of the El Niño Southern Oscillation (ENSO).  So when referring to the climate pattern in general (e.g. line 436) use ENSO.  When referring to the warm phase (line 437, 438), use El Niño instead.  And use this notation consistently throughout the text.

*Thank you for this suggestion. We have made the appropriate changes in the text and clarified the difference between ENSO and El Niño in the following adapted sentence in the introduction: "Finally, hot and dry conditions associated with the warm phase (El Niño) of the El Niño Southern Oscillation (ENSO) and ... " L106*

L458.  Although there is a large variability in drought strategies in the Amazon, and there are trees that can keep transpiration at similar levels during dry periods (Maréchaux et al. 2018; Oliveira et al. 2019).

*Yes, we agree. We have added this nuance at the end of the first paragraph of the introduction: "Although there is a large variability of drought avoidance and drought tolerance strategies between Amazonian tree species, with some trees maintaining transpiration and leaf area during drought (Bonal et al., 2000a; Brum et al., 2018, 2019; Janssen et al., 2020, 2019; Maréchaux et al., 2018; Oliveira et al., 2019), if a proportion of trees sheds their leaves to avoid dehydration, all other things being equal, this will show up as increased leaf litterfall on the stand scale."*

L479. "Attributed" instead of "contributed".

*Thank you, this has been changed.*

L516. Could drought duration have played a role here?

*Yes we believe this has to do with drought duration, timing and intensity, we have included this now in the text.*

Table 1. I think the table should include all the data used in the XGBoost models, including the remote sensing and the derived quantities. Also, indicate which version of SoilGrids was used, and which soil layers were used (or if all depths were used). For ERA5, there is now a peer-reviewed publication too (Hersbach et al. 2020).

*Thank you for this suggestion. The remote sensing data was not used in the XGBoost models so it is not included in Table 1 to avoid confusion.*

*The following sentence was added to the caption of Table 1: "The SoilGrids dataset (Hengl et al., 2017) contains data from seven soil layers at different depths below the surface. For this study, these layers were merged into two layers with a shallow soil layer (L1-L3) and a deep soil layer (L4-L7)."*

*The reference to the Hersbach et al. (2020) paper has now been included in the manuscript, replacing the former Copernicus and ECMWF references.*

Figures 2–4, 6–7. I suggest using more intuitive palettes (viridis, magma for the absolute plots, divergent palettes with white near zero for anomalies). Also red-green scales can be difficult for some people.

*Thank you for the suggestion. We now use the viridis pallet for the absolute plot (Figure 2) and the plot with uncertainty estimates (Figure S4). We believe that the pallet for anomalies is appropriate and intuitive as negative anomalies are shown as red and positive as green, with white near zero.*

**References**

Aragão, L. E. O. C., et al.: 21st Century drought-related fires counteract the decline of Amazon deforestation carbon emissions, Nature Comm., 9, 536, doi:10.1038/s41467-017-02771-y, 2018.

Chave, J., et al.: Error propagation and scaling for tropical forest biomass estimates, Philos. Trans. R. Soc. B-Biol. Sci., 359, 409–420, doi:10.1098/rstb.2003.1425, 2004.

Gloor, M., et al.: Intensification of the Amazon hydrological cycle over the last two decades, Geophys. Res. Lett., 40, 1729–1733, doi:10.1002/grl.50377, 2013.

Hansen, M. C., et al.: High-Resolution Global Maps of 21st-Century Forest Cover Change, Science, 342, 850–853, doi:10.1126/science.1244693, 2013.

Hersbach, H., et al.: The ERA5 Global Reanalysis, Quart. J. Royal Meteorol. Soc., 146, 1999–2049, doi:10.1002/qj.3803, 2020.

Hirota, M., et al.: Global Resilience of Tropical Forest and Savanna to Critical Transitions, Science, 334, 232–235, doi:10.1126/science.1210657, 2011.

Longo, M., et al.: Ecosystem heterogeneity and diversity mitigate Amazon forest resilience to frequent extreme droughts, New Phytol., 219, 914–931, doi:10.1111/nph.15185, 2018.

Maréchaux, I., et al.: Dry-season decline in tree sapflux is correlated with leaf turgor loss point in a tropical rainforest, Funct. Ecol., 32, 2285–2297, doi:10.1111/1365-2435.13188, 2018.

Morton, D. C., et al.: Amazon forests maintain consistent canopy structure and greenness during the dry season, Nature, 506, 221–224, doi:10.1038/nature13006, 2014.

Oliveira, R. S., et al.: Embolism resistance drives the distribution of Amazonian rainforest tree species along hydro-topographic gradients, New Phytol., 221, 1457–1465, doi:10.1111/nph.15463, 2019.

Song, X.-P., et al.: Global land change from 1982 to 2016, Nature, 560, 639–643, doi:10.1038/s41586-018-0411-9, 2018.

Tomasella, J., et al.: The droughts of 1996-1997 and 2004-2005 in Amazonia: hydrological response in the river main-stem, Hydrol. Process., 25, 1228–1242, doi:10.1002/hyp.7889, 2011.

Withey, K., et al.: Quantifying immediate carbon emissions from El Nio-mediated wildfires in humid tropical forests, Philos. Trans. R. Soc. B-Biol. Sci., 373, 20170 312, doi:10.1098/rstb.2017.0312, 2018.

**Reply to referee #2**

**General comments**

Janssen et al's manuscript brings a very good contribution to understanding tropical forest responses to multidecadal climatic variation across the Amazon basin by combining available in situ measurements of components of forest productivity (long-term records of stem growth and leaf litterfall) and remote sensing products under a robust modelling approach. Results show significant long-term trends of decreasing stem growth, and a less strong increase of leaf litterfall in the Amazon basin since the early 1980s.

Overall, the manuscript is very well written, and I have no issues on number and quality of figures/ tables, or language. I have some specific questions on the methods as described below, and I hope solving these questions will improve the manuscript.

*Thank you very much for your kind review. The time and effort put into the reading of our manuscript and writing this review is greatly appreciated. Furthermore, thank you for recognizing our work as a potential useful contribution to the existing literature. Below, we will address your concerns and questions point by point.*

Although the authors compiled data from the literature from sites across the neotropics, the evaluated effects of drought on modelled leaf fall, leaf flushing and stem growth are centered in the Amazon basin; therefore, I believe the manuscript title is misleading the readers about drought effects on "Neotropical forest". Please consider this aspect.

*We agree that while our dataset covers the entire neo-tropics, the analysis is focused specifically on the Amazon basin. Therefore, we changed the title into: "Drought effects on leaf fall, leaf flushing and stem growth in the Amazon forest; reconciling remote sensing data and field observations". We now also specified the region of interest (the Amazon basin) in the aims of the study in the final paragraph of the Introduction (L111-L116).*

**Specific comments**

The observational dataset on stem growth and litterfall was not available for review, but I think it is valid to describe how data on stem growth and litterfall are distributed along the period 1982-2019; is it uniform? If not (more data on the first two decades when compared to more recent years), how did this discrepancy (or unbalance) affect the predicted values? Was that issue contemplated during the machine learning process (L230-243)?

*Considering the imbalance of leaf litterfall and stem growth observations over time in the dataset, it is true that the density of observations changes over time and that this might*

*impact the model uncertainties as well. We now show these trends in a new supplementary figure (Figure SX). In general, there was an increase in observations of both leaf litterfall and stem growth in the 2000's compared to the 1980's and 1990's (Figure SX). However, since 2010 there has been a steady decline in the number of observations per year which can at least partly be explained by a larger contribution of data that is presently not published or under embargo. There is also a positive trend in the normalized root mean squared error (RMSE) of the leaf litterfall and stem growth model estimates (Figure SX) suggesting that the relative model error increases over time in both models. For stem growth, the increasing trend in NRMSE is mainly driven by the decline of the averaged predicted stem growth in the dataset while the absolute error did not change over time (Figure 1c). In the case of leaf litterfall, the increase in NRMSE over time is driven by the increase in absolute error (Figure 1c) despite the increase of average leaf litterfall over time. This suggests that despite more data being available in recent years, the model error is actually higher compared to the 1980's and 1990's. We acknowledge that these issues were not specifically discussed in our manuscript. In the revised manuscript we examine these uncertainties in more detail. Furthermore, the dataset compiled for this study will become public in the final stages of the publication process.*

Observational data was compiled for the entire Neotropics (as shown in Fig 2a, b, and described in the methods), so why did the subsequent core analyses have "cropped" a specific area centered in the Amazon basin (15oN – 20oS), excluding central America, Atlantic forest and part of the Brazilian caatinga and Cerrado? I could not find any explanation in the methods on this topic.

*We have "cropped" the model output and remote sensing images for the analyses to cover the extent of the Amazon basin. This was done because the focus of the study was on the drought related changes in forest growth that occurred in the Amazon basin. Furthermore, the figures were cropped to cover the area of interest as local differences cannot be identified if the entire study area was shown (details are lost when "zooming out"). Furthermore, as can be observed in for example Figure 3 and briefly discussed in the results (L345-348), different responses in the modelled stem growth and leaf litterfall and the remote sensing data are shown for the humid forests of the Amazon basin and the more dry Cerrado and caatinga regions. It would have made the paper considerably lengthier if we had to examine the drought-induced changes that occurred in every region or ecosystem covered by the model output. We have now specified in the aims of the study (Introduction, final paragraph) that we focus our analysis on the Amazon basin.*

Figure 8: What is the interpretation of the authors on the seasonal anomalies (especially Fig. 8c) shown in middle 2006? Those values look significantly higher than the ones found for the 2005 drought period (gray area in the figure).

*Thank you for noting the significant negative anomaly in soil moisture and positive anomaly in VPD in April 2006 that can be observed in Figure 8 of the manuscript and the associated negative anomalies in stem growth in the same period. We believe that this can be considered a short drought period, which indeed seems more anomalously dry compared to the 2005 drought. This short drought period is now mentioned in the results section of the revised manuscript (section 3.4, 4th and 6th paragraph).*

On the conclusion (L580-583), it is unclear what is the opinion of the authors on the inconsistencies between the trends obtained here using modelled time series associated plus remote sensing and the trends observed using long-term inventory plots. Are remote sensing and ground-based trends reconcilable? Plot networks are not free from spatial bias, as correctly pointed out by the authors, so are plot-based carbon sequestration overestimated?

*We agree that in the conclusion we do not reveal our opinions about whether remote sensing and ground-based trends can be reconciled and whether ground based carbon sequestration is overestimated. What we aimed to communicate in the discussion is that remote sensing techniques differ considerably in what they measure and how these measurements are related to forest growth and "health". For example, we show that canopy greening is not a good proxy of above-ground growth or drought stress in evergreen tropical forests. Furthermore, we show that ground-based estimates can be upscaled using climate data and other geospatial datasets to obtain a different temporal trend in stem growth compared to when the measurements of largely the same inventory plots are averaged. It would be interesting for future research to examine which upscaling technique provides the most accurate trend estimation, but for now we cannot confidently conclude which technique is more appropriate. We have expanded our discussion about the discrepancy in the discussion section (section 4.2, third paragraph), also after comments from referee #1.*

L583-585: actually, tree mortality rates can be even higher few years after drought (not during the droughts) as drought-induced mortality is not instantaneous

*Yes, this sentence has been adapted to: "As tree mortality is elevated during and following drought (Feldpausch et al., 2016; Phillips et al., 2009) ..."*

**Technical corrections**

L301-304: How did "biomass production" was estimated in Fig 2c and Fig 9c?

*We have now included the definition of above-ground biomass production in the text. The sentence was changed to: "As the range in predicted leaf litterfall rates was much larger than the range in predicted stem growth rates, the spatial variability in leaf litterfall rates largely drives the spatial variability in aboveground biomass production (defined as the multi-year sum of leaf litterfall and stem growth) across the Neotropical ecosystems (Figure 2c)."*

L345-348: some readers may not know where Cerrado and Caaating regions are located in the map; please use geo-locators, marks, or arrows to point out those areas

*Thank you for the suggestion. We agree that some readers might not know where the Cerrado and Caatinga regions are located. We have adapted the sentence to: "Note the contrast in the observed responses between the moist tropical forest of the Amazon basin (inside the black contour line) with the Cerrado and Caatinga regions, located to the south and south-east of the Amazon basin of in eastern Brazil." We chose to highlight the location in the text and not in the Figure as we believe this might distract the reader from what is happening inside the Amazon basin, within the black contour, which is the focus of the study. We hope this will be sufficient for all readers.*

Figure 8 was mentioned in the text before figures 6 and 7; please review

*Thank you for noticing this, the entire section has been restructured so that the results presented in Figure 6 and 7 are now reported before the results in Figure 8.*

Fig 9: please provide a source (reference) for the multivariate ENSON index (or include it Table 1)

*A reference is included in the Results section (L566) and in the figure Caption.*

Please, check if all references are correctly cited according to the Journal's recommendation; some of them looks incomplete, for instance, L972

*Thank you for noticing, the reference list was checked for other errors.*

P39, L960: hyperlink for the publication is broken, it looks like missing letter "r" in the end

*Thank you for noticing, the "r" was indeed missing from the hyperlink. This has now been fixed.*